# The interdomain flexible linker of the polypeptide GalNAc transferases dictates their long-range glycosylation preferences

Matilde de las Rivas[1], Erandi Lira-Navarrete[1,2], Earnest James Paul Daniel[3], Ismael Compañón[4], Helena Coelho[5,6,7], Ana Diniz[5], Jesús Jiménez-Barbero[6,7,8], Jesús M. Peregrina[4], Henrik Clausen[2], Francisco Corzana [4], Filipa Marcelo[5], Gonzalo Jiménez-Osés[4], Thomas A. Gerken[3,9,10] & Ramon Hurtado-Guerrero[1,11]

The polypeptide GalNAc-transferases (GalNAc-Ts), that initiate mucin-type O-glycosylation, consist of a catalytic and a lectin domain connected by a flexible linker. In addition to recognizing polypeptide sequence, the GalNAc-Ts exhibit unique long-range N- and/or C-terminal prior glycosylation (GalNAc-O-Ser/Thr) preferences modulated by the lectin domain. Here we report studies on GalNAc-T4 that reveal the origins of its unique N-terminal long-range glycopeptide specificity, which is the opposite of GalNAc-T2. The GalNAc-T4 structure bound to a monoglycopeptide shows that the GalNAc-binding site of its lectin domain is rotated relative to the homologous GalNAc-T2 structure, explaining their different long-range preferences. Kinetics and molecular dynamics simulations on several GalNAc-T2 flexible linker constructs show altered remote prior glycosylation preferences, confirming that the flexible linker dictates the rotation of the lectin domain, thus modulating the GalNAc-Ts' long-range preferences. This work for the first time provides the structural basis for the different remote prior glycosylation preferences of the GalNAc-Ts.

[1] BIFI, University of Zaragoza, BIFI-IQFR (CSIC) Joint Unit, Mariano Esquillor s/n, Campus Rio Ebro, Edificio I+D, Zaragoza, 50018, Spain. [2] Copenhagen Center for Glycomics, Department of Cellular and Molecular Medicine, School of Dentistry, University of Copenhagen, Copenhagen, DK-2200, Denmark. [3] Department of Biochemistry, Case Western Reserve University, Cleveland, 44106 OH, USA. [4] Departamento de Química, Universidad de La Rioja, Centro de Investigación en Síntesis Química, E-26006 Logroño, Spain. [5] UCIBIO, REQUIMTE, Departamento de Química, Faculdade de Ciências e Tecnologia, Universidade de Nova de Lisboa, Caparica, 2829-516, Portugal. [6] CIC bioGUNE, Bizkaia Technology Park, Building 801A, 48170 Derio, Spain. [7] Department of Organic Chemistry II, Faculty of Science & Technology, University of the Basque Country, Leioa, Bizkaia 48940, Spain. [8] Ikerbasque, Basque Foundation for Science, Maria Diaz de Haro 13, 48009 Bilbao, Spain. [9] Department of Pediatrics, Case Western Reserve University, Cleveland, 44106 OH, USA. [10] Department of Chemistry, Case Western Reserve University, Cleveland, 44106 OH, USA. [11] Fundación ARAID, 50018 Zaragoza, Spain. Matilde de las Rivas, Erandi Lira-Navarrete and Earnest James Paul Daniel contributed equally to this work. Correspondence and requests for materials should be addressed to R.H-G. (email: rhurtado@bifi.es)

Polypeptide *N*-acetylgalactosaminyltransferases (GalNAc-Ts) are higher eukaryotic-retaining glycosyltransferases (GTs) that transfer a GalNAc moiety from uridine diphosphate *N*-acetylgalactosamine (UDP-GalNAc) onto Ser/Thr residues of proteins[1]. This family of enzymes, which initiates the most common type of *O*-glycosylation in metazoans[1–5], plays profound roles for the structure, stability, and function of proteins[6]. Gal-NAc-*O*-glycosylation is one of the most abundant types of protein glycosylation in eukaryotes that is chiefly found in the densely clustered heavily glycosylated domains of mucins and mucin-domain-containing glycoproteins, hence the name, mucin-type *O*-glycosylation. However, it is also clear that many other proteins contain isolated sites of GalNAc-*O*-glycosylation, where more than 5000 human proteins trafficking the secretory pathway have been identified to date containing one or more mucin-type *O*-glycans[7].

The initiation of mucin-type *O*-glycosylation by the GalNAc-Ts is one of the most complex regulated types of protein glycosylation. The GalNAc-Ts comprise a family of up to 20 iso-enzymes possessing a range of kinetic properties and expression patterns. These enzymes orchestrate with high fidelity, the initial patterns of *O*-glycosylation on diverse proteins, including the high-density regions in mucins, where 30–50% of the amino acids may be glycosylated[1, 8]. The GalNAc-Ts are unique among metazoan GT enzymes because in addition to their N-terminal catalytic domain adopting a GT-A fold (family 27 in the CAZy database[9]), they possess a C-terminal GalNAc-binding lectin domain with a β-trefoil fold[4, 10–13] (classified as a carbohydrate-binding module 13 in the CAZy database), which provides additional functions to these enzymes[14] (see below for further details). Both domains are linked through a short flexible linker whose motion has been suggested to be responsible for the dynamic conformational landscape of these enzymes[4].

GalNAc-Ts can be classified according to their different glycosylation capacity on peptide and glycopeptide substrates. Three distinctive glycosylation modes have been reported: catalytic domain-dependent glycosylation on "naked" peptides (Fig. 1a) and on glycopeptides (Fig. 1b), hereafter termed short-range glycosylation[15, 16]; and remote lectin domain-dependent

glycosylation on glycopeptides (Fig. 1c), termed long-range glycosylation[15, 16]. Adjacent and proximal sites relative to an existing GalNAc glycosite are glycosylated in a catalytic domain-dependent manner (Fig. 1), with GalNAc-T4 being part of a small number of GalNAc-Ts (including GalNAc-T7 and T10) that will also glycosylate contiguous sites[16]. Distant sites can be glycosylated in a lectin domain-dependent manner (Fig. 1) with individual GalNAc-Ts having distinct preferences on remote sites located N or C terminus from a prior GalNAc glycosite[15, 16]. For example, GalNAc-T1/T2/T14 tend to glycosylate preferentially N-terminal sites remote from prior C-terminal GalNAc glycosites in glycopeptides, whereas GalNAc-T3/T4/T6/T12 exhibit the opposite preference (Fig. 1c). Furthermore, some GalNAc-Ts, including GalNAc-T5/T13/T16, appear to exhibit both orientational preferences for long-range glycosylation[16]. Previous structural work with GalNAc-T2 in complex with "naked" peptides and glycopeptides has provided insight into the catalytic domain-dependent reaction, and also demonstrated how the GalNAc-T2 lectin domain guides catalysis to N-terminal acceptor sites very distant from a prior C-terminal GalNAc glycosite[4, 12, 17]. However, the molecular basis of how other GalNAc-Ts, such as GalNAc-T3/T4/T6/T12, achieve the opposite long-range glycosylation preferences remains unclear[15, 16].

The GalNAc-T4 isoform is of particular interest as it is the only isoform capable of glycosylating two out of the five acceptor sites in the partially glycosylated MUC1 mucin tandem repeat in a lectin domain-dependent manner[18]. The density of glycosylation together with the structure of *O*-glycans in the mucin repeat regions are important features for the cancer-glycoforms of MUC1 being exploited for diagnostic as well as therapeutic purposes[19]. Generally, GalNAc-T4 glycosylates very few isolated glycosylation sites in "naked" peptide acceptors, although it readily glycosylates the important Thr57 in PSGL-1 that is the essential *O*-glycan required for P-selectin-mediated leukocyte trafficking[20, 21]. We report herein a multidisciplinary approach on GalNAc-T4, combined with the characterization of different chimeras and mutants of GalNAc-T2, which begins to reveal the molecular basis of the long-range glycosylation preferences of the GalNAc-Ts on glycopeptide acceptors substrates.

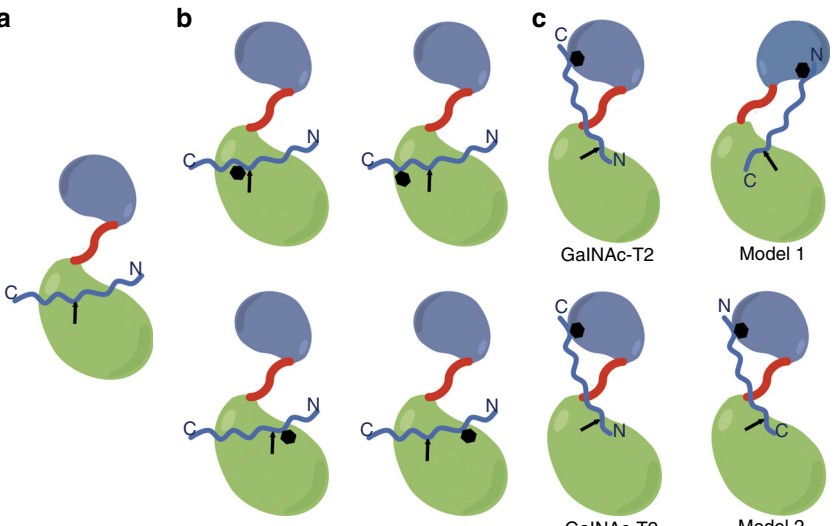

**Fig. 1** Modes of *O*-glycosylation found for the GalNAc-Ts. **a–c** These panels denote the three distinct modes of *O*-glycosylation performed by GalNAc-Ts: the neighboring glycosylation activity by the catalytic domain in **a** and **b**, and long-range lectin domain-mediated glycosylation activity in **c**. In **c**, two plausible models are suggested that might explain the different long-distance glycosylation preferences of GTs such as GalNAc-T3/T4/T16/T12. Oval-shaped figures in blue and green depict the lectin and catalytic domains, respectively. Peptides are indicated in blue and the black hexagon-shaped figure denotes the position of prior GalNAc moieties in the glycopeptides. Arrows indicate the positions of acceptor sites

## Results

**Models explaining the long-range glycosylation preferences.**
Based on previous results, we can infer two possible models (see Fig. 1c) that might provide a plausible explanation for the Gal-NAc-T3/T4/T6/T12 long-range glycosylation preference in contrast to the one found for enzymes such as GalNAc-T1/T2/T14[15, 16]. In model 1, the rotation of the lectin domain GalNAc-binding site is required, thereby maintaining the same orientation of the peptide on the catalytic domain. In model 2, the lectin domain GalNAc-binding site would adopt the same orientation as found in GalNAc-T2 while the glycopeptide adopts an inverse orientation on the catalytic domain, an assumption unlikely to happen providing the high similarity of the peptide-binding groove among GalNAc-Ts[11] (Fig. 1).

**GalNAc-T4-substrate interaction and kinetics studies.** To test the two models proposed above, three simplified peptides enriched in glycine and alanine residues were designed, synthetized, and evaluated as substrates of GalNAc-T4 (Table 1). One of these peptides was a "naked" peptide, **1** (denoted −TT− for simplicity), whereas the other two were monoglycopeptides, **2** and **3** (denoted -TT–T*- and -T*–TT-, where T* represents a GalNAc glycosylated Thr), that contain a GalNAc moiety located either at the C or N terminus, respectively. All peptides have two potential Thr acceptor sites and also contain an adjacent PXP motif, which is recognized by most GalNAc-Ts[4, 16]. Our kinetic studies showed that GalNAc-T4 glycosylated selectively peptide **3** (with a high affinity, $K_m \sim 40\,\mu M$, and catalytic efficiency, ~75 (nmole min)$^{-1}$) over peptides **1** and **2**, both of which were imperceptibly glycosylated over a range of peptide concentrations (Fig. 2a). This is consistent with previous findings[16] on GalNAc-T4 and illustrates that the lectin domain guides GalNAc-T4 to preferentially glycosylate C-terminal sites of monoglycopeptides that are remote from an N terminus GalNAc-glycosylated residue. Although, the "naked" peptide **1** was not a substrate, GalNAc-T4 has been reported to readily glycosylate other highly specific peptide sequences[20, 21].

### Table 1 Peptide acceptor substrates used in this study

| Peptide | Sequence |
| --- | --- |
| **1** (−TT−) | GAGAGAGTTPGPG |
| **2** (-TT–T*-) | AGAGTTPGPGAGAT*GA |
| **3** (-T*–TT-) | GAT*GAGAGAGTTPGPG |

T* denotes the Thr-O-GalNAc moiety

To understand why GalNAc-T4-only glycosylated monoglyco-peptide **3**, we performed binding studies using saturation-transfer difference (STD) NMR experiments on peptide substrates **1–3** and α-methyl-GalNAc in the presence of UDP and MnCl$_2$ (Fig. 2b and Supplementary Fig. 1). The absence of STD-NMR signals for peptide **1** revealed that the "naked" peptide was very poorly recognized by GalNAc-T4 in agreement with the low glycosylation observed during the kinetics assays. For both monoglycopeptides **2** and **3** relatively weak STD-NMR signals were observed for the Ala and Thr methyl protons, suggesting weak recognition of the peptide by GalNAc-T4. However, the O-GalNAc glycosylated Thr of the monoglycopeptides **2** and **3** receive a higher STD enhancement. Remarkably, the sugar protons of the GalNAc moiety of the monoglycopeptides **2** and **3** (region 3.5 a 4.2 ppm of the spectrum together with the singlet at 2 ppm) showed unambiguous and more intense STD-NMR signals (Fig. 2b and Supplementary Fig. 1). In addition, the STD-NMR-derived epitope mapping of the GalNAc moiety is identical for both monoglycopeptide **2** and control α-methyl-GalNAc. In fact, for both compounds, the highest STD response corresponded to the H2 proton of the GalNAc moiety followed by a modest STD of protons H4, H3, and N-acetyl group, suggesting similar lectin-binding modes for these molecules. In contrast, monoglycopeptide **3** presents a somewhat different GalNAc STD-NMR-binding pattern, where H2, H4, and the N-acetyl group of the GalNAc moiety have the highest STD-NMR response closely followed by the H3 proton. This suggests that the monoglycopeptide **3** has a distinct binding mode of interaction with the enzyme, which is different to the one found for monoglycopeptide **2** and α-methyl-GalNAc control. From our kinetic and STD-NMR results, we can infer that the unique binding of the GalNAc residue of monoglycopeptide **3** to the lectin domain, facilitates the correct orientation of the peptide acceptor onto the catalytic domain for optimal C-terminal glycosylation. Thus, even though monoglycopeptide **2** shows GalNAc binding to the lectin domain, its N-terminal acceptor site must be directed away from the catalytic domain, leading to the inability of GalNAc-T4 to glycosylate this substrate. Hence, the correct location of a prior glycosite in a monoglycopeptide, with respect to potential acceptor sites, is essential for an optimal interaction of the peptide within the catalytic domain, which in turn is critical for an efficient catalysis and specificity.

**Crystal structure of GalNAc-T4.** To further elucidate the molecular basis of GalNAc-T4's monoglycopeptide glycosylation preference and rule out one of the two proposed models (Fig. 1),

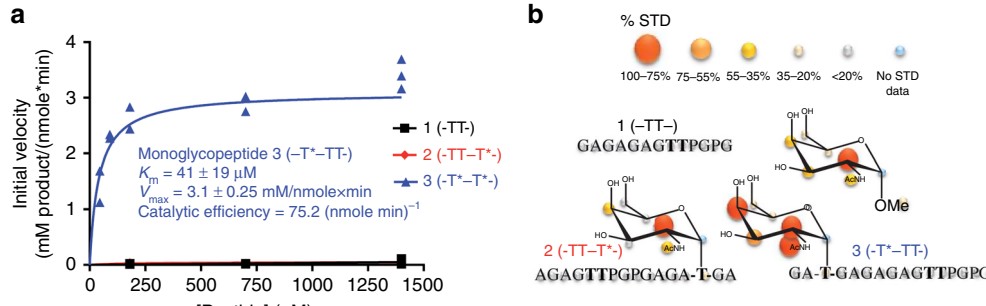

**Fig. 2** Biophysical characterization of GalNAc-T4 against peptides **1–3**. **a** Peptide glycosylation kinetics of GalNAc-T4 against (glyco)peptides **1–3** (black, red, and blue symbols, respectively). Michaelis–Menten kinetic values, $K_m$, $V_{max}$, and catalytic efficiency ($V_{max}/K_m$) for monoglycopeptide **3** were obtained from the nonlinear least square fit to the initial rate data, obtained as described in the "Methods" section and given in Supplementary Table 2. Peptide substrates **1** and **2** are largely unglycosylated by GalNAc-T4. **b** STD-NMR-derived epitope mapping. The different colored spheres indicate the normalized STD signal (in percent) observed for each proton. See "Methods" section and Supplementary Fig. 1 for further details

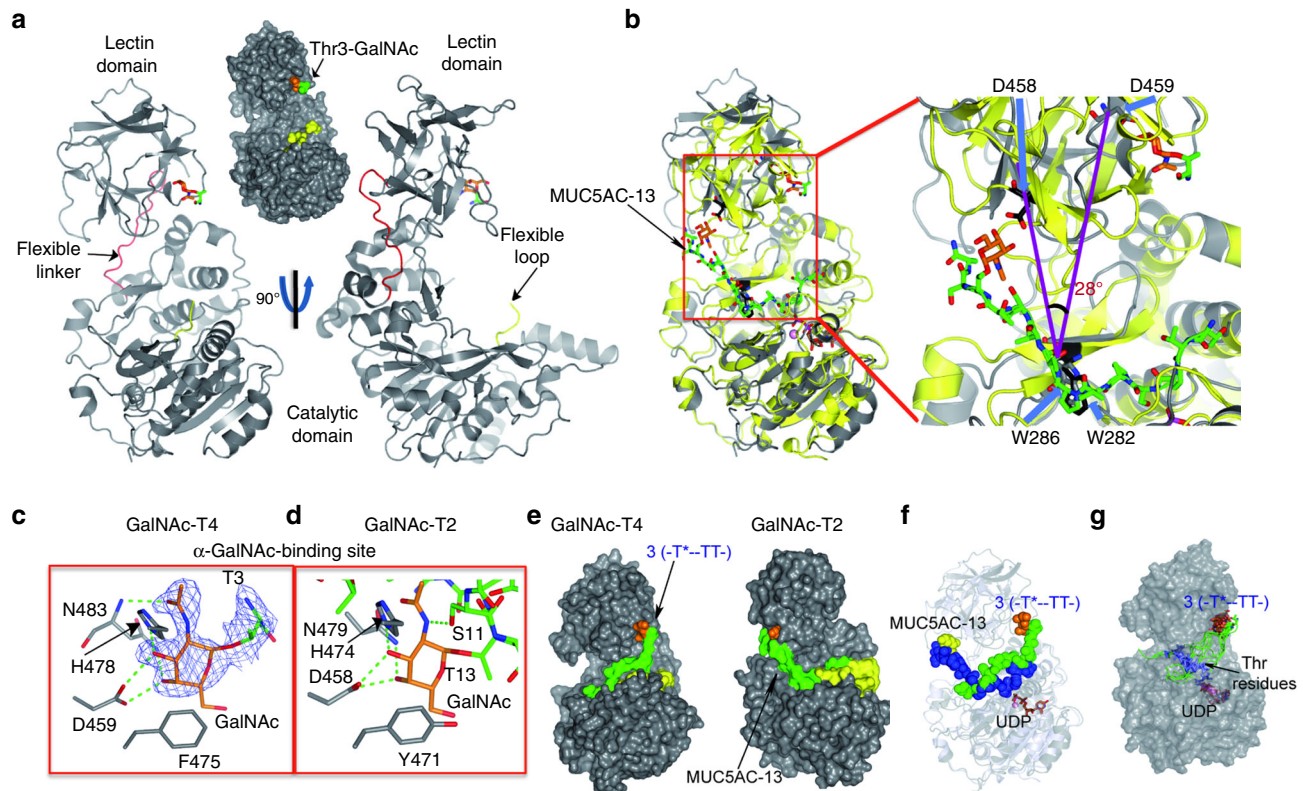

**Fig. 3** Structural characterization of the interaction between GalNAc-T4 and monoglycopeptide 3. **a** Two different views of GalNAc-T4 in complex with 3. The catalytic and lectin domains are colored in gray and the interdomain flexible linker is depicted in red. The catalytic domain flexible loop is depicted in yellow and is mostly disordered. The GlcNAc moiety of the Thr3-GalNAc is shown as orange carbon atoms while Thr3 is shown as green carbon atoms. **b** (left) Superposition of the GalNAc-T4-glycopeptide 3 (gray) and GalNAc-T2-MUC5AC-3-UDP-Mn$^{+2}$ (yellow) structures. The glycopeptide and the GalNAc moiety are shown in green and orange carbon atoms, respectively. The lectin α-subdomain GalNAc-binding residues, Asp458, and Asp459, of GalNAc-T2 and GalNAc-T4 are shown as sticks in black and gray carbon atoms, respectively. (right panel) Close-up view of the superposition between GalNAc-T2 and GalNAc-T4. Colors for peptides, GalNAc moiety, and proteins are identical as shown above. **c–d** Close-up view of the lectin α-subdomain GalNAc-binding site for both GalNAc-T4 (left) and T2 (right). The residues of both enzymes are depicted as gray carbon atoms. Hydrogen bond interactions are shown as dotted green lines. Electron density maps are F$_O$–F$_C$ (blue) contoured at 2.2 σ for Thr3-GalNAc. Note that both GalNAc-binding sites are depicted in the same orientation for comparison. **e** Surface representation of GalNAc-T4 (model built with monoglycopeptide 3 and UDP/Mn$^{+2}$), and GalNAc-T2 (with UDP/Mn$^{+2}$/MUC5AC-13). Both are viewed from the same orientation as in **b**. Colors for the glycopeptide and the flexible loop are the same as above. **f** Superposition of the GalNAc-T4-glycopeptide 3-UDP-Mn$^{+2}$ (gray) and GalNAc-T2-MUC5AC-3-UDP-Mn$^{+2}$ (blue–white) structures. The glycopeptide and the GalNAc moiety in the structure of GalNAc-T2 and GalNAc-T4 are shown as spheres in blue/green and yellow/orange, respectively. UDP and Mn$^{+2}$ are shown as brown carbon atoms and as a pink sphere, respectively. **g** Superimposed structures of the 200 ns MD simulation trajectory for GalNAc-T4 in complex with glycopeptide 3/UDP/Mn$^{+2}$. GalNAc-T4 is depicted in a light gray surface view. The GalNAc moiety, the peptide backbone, UDP, and the acceptor Thr residues are shown as orange, green, brown, and blue carbon atoms, respectively

we obtained triclinic crystals of GalNAc-T4 that were subsequently soaked with monoglycopeptide 3 and UDP and MnCl$_2$. The resulting crystal allowed us to solve the corresponding structure at a high resolution (1.90 Å) (Supplementary Table 1). Within the asymmetric unit, two independent GalNAc-T4 molecules were present. The crystal structure shows a compact structure with the typical GT-A fold and the lectin domain located at the N- and C-terminal regions, respectively (Fig. 3a). A sequence alignment between GalNAc-T2 and T4 displays a sequence identity of ~40% and a high resemblance between these two enzymes at the secondary structure level (Supplementary Fig. 2). In fact, superposition analysis of both enzymes shows that both catalytic domains are more structurally similar (root-mean-square deviation (RMSD) of 1.23 Å for aligned Cα atoms) than the lectin domains (RMSD of 1.82 Å) (Fig. 3b and Supplementary Fig. 3). A global superposition of both enzymes renders a larger RMSD of 2.12 Å, which suggests that a major shift has occurred between the orientation of the lectin domains (see below for a further discussion; Fig. 3b). As expected and accordingly to model 1, the GalNAc-binding site in GalNAc-T4 is located on the right

side of the lectin domain, which requires a rotation of 28° with respect to the homologous site in GalNAc-T2 (Fig. 3b and Supplementary Fig. 4). To calculate the angle of rotation, the Cαs of Asp458, Trp282, and Asp459 were used. Note that the angle of rotation could be also determined using the Cαs of Asp458, Trp286, and Asp459. In this case, the rotation angle was 30°, which is very close to 28°. This is due to Trp282 of GalNAc-T2 and Trp286 of GalNAc-T4 adopt almost identical positions in the catalytic domain (Fig. 3b). This different position of the GalNAc-T4 GalNAc-binding site, consistent with the STD-NMR results, may explain why GalNAc-T4 promotes a distinct type of long-range glycosylation, consisting of the glycosylation of acceptor sites located at the C-terminal from prior GalNAc glycosites in glycopeptides. This also suggests that the location of the GalNAc-binding site of the lectin domain is coupled with the long-range glycosylation preferences of these enzymes.

**Lectin domain-binding site of GalNAc-T4.** Even though the crystals were soaked with a high concentration of peptide 3, UDP

and MnCl$_2$, we could only visualize well-defined density for the GalNAc-$O$-Thr moiety of monoglycopeptide **3** bound to the lectin α-subdomain GalNAc-binding site (Fig. 3a, c). Note that GalNAc-Ts lectin domains contain three potential carbohydrate-binding subdomains named α, β, and γ; however, in most GalNAc-Ts only one of them is functional[15, 16, 18, 22, 23]. The almost absence of density for the peptide backbone of mono-glycopeptide **3**, is consistent with the STD-NMR results (showing no significant STD with the acceptor peptide residues) and with the weak binding affinity of this peptide ($K_d$ estimated in the mM range), determined by surface plasmon resonance (SPR) experiments in the presence of UDP and MnCl$_2$ (Supplementary Fig. 5). The discrepancies between GalNAc-T4's high-affinity $K_m$ obtained from our kinetics studies (Fig. 2a) versus the poor $K_d$ values from the binding studies can be attributed to the fact that the enzyme kinetics was performed with UDP-GalNAc. It is known that this donor substrate stabilizes the flexible loop of the catalytic domain active site in a closed conformation, thus completing the formation of the peptide-binding groove and leading to an active enzyme[4, 12]. Thus, our observations in the absence of UDP-GalNAc represent the weak binding of peptide substrate to the open flexible loop conformation of the enzyme.

A closer inspection at the lectin domain GalNAc-binding site depicts that the GalNAc moiety was tethered by conserved residues, including Asp459, Asn483, Phe475, and His478, of the lectin α-subdomain GalNAc-binding site (equivalent residues in GalNAc-T2 are Asp458, Asn479, Tyr471, and His474; Fig. 3d). It is important to note that only the lectin α-subdomain GalNAc-binding site has been shown to be functional in both GalNAc-T2 and GalNAc-T4 lectin domains[4, 18]. Most of the interactions are through hydrogen bonds in both enzymes, although they also share CH–π interactions between the GalNAc moiety and Phe475/Tyr471. The importance of the Asp residue in recognition of the GalNAc moiety and its role in the lectin domain-mediated glycosylation has been previously reported[4, 18]. Furthermore, the crystal structure explained the GalNAc STD-NMR-derived epitope mapping. In particular, the structure shows that H4 and H3 of the GalNAc moiety are in closer contact with Phe475 and His478, which is further supported by the STD response (92% for H4 and 72% for H3) observed for these protons in monoglycopeptide **3**.

**Molecular docking and dynamics simulations**. To address how the lectin domain guides the delivery of potential peptide acceptor sites to the catalytic domain, we performed molecular docking and dynamics (MD) simulations on GalNAc-T4 in complex with UDP/Mn$^{+2}$ and monoglycopeptide **3** (see "Methods" section). The calculations suggest that although the peptide bound at the catalytic domain was highly dynamic, the two potential peptide acceptor Thr residues were in close contact to the β-phosphate of UDP during the simulated timeframe (200 ns), thereby accounting for the glycosylation of this peptide (Fig. 3e, f, g and Supplementary Fig. 6 and Supplementary Movie 1). A comparison with the GalNAc-T2 structure in complex with MUC5AC-13 and UDP/Mn$^{+2}$ clearly provides evidence of the two different orientations of the lectin domains and supports our conclusion on the coupling of the location of the GalNAc-binding site with the observed long-distance glycosylation preference (Fig. 3e, f).

**The flexible linker dictates the lectin domain rotation**. Previous studies suggested that the flexible linker allowed for the inter-domain translational-like motion in these enzymes[4]. We therefore rationalized that different flexible linkers found in these GTs might also be behind the rotation capacity of the lectin domain (Supplementary Fig. 7). To test this hypothesis, we initially performed 500 ns MD simulations in water solution of native GalNAc-T2 and GalNAc-T4. Simulations clearly showed that the lectin domains of GalNAc-T2 and GalNAc-T4 *apo* enzymes do not rotate in the considered timescale (Supplementary Movies 2 and 3). This suggests that the orientation of the GalNAc-binding site with respect to the catalytic domain may remain relatively fixed in a fully folded transferase. We then performed MD simulations on **chimeras 1** and **2** that corresponded to GalNAc-T2 with two different lengths of the flexible linker from GalNAc-T3 (Table 2). In agreement with our hypothesis, we observed the prompt rotation of the lectin domain of the GalNAc-T2 **chimera 2** toward a position similar to that in GalNAc-T4 (Fig. 4a, Supplementary Fig. 8, and Supplementary Movie 4). This stepwise motion was completed in ~30 ns and involved sequential extension of the flexible linker, rotation of the lectin domain around the $Z$ and $Y$ axes, and finally compression of the linker (Fig. 4a). Thus, initially the GalNAc-T3 linker was artificially forced to adopt the compact conformation of the GalNAc-T2 native linker, but early on in the simulation (after ~10 ns), the linker quickly recovered its more native extended conformation and as a consequence, the lectin domain springs and rotates sidewise to finally re-assemble in a structure similar to that of our GalNAc-T4. This motion is likely smoothed by the absence of strong interdomain

---

**Table 2 Sequences of the chimeras and mutant GalNAc-Ts used in this study**

| Transferase | Sequence of flexible linker[a] |
|---|---|
| GalNAc-T2 | P$_{435}$ELRVPDHQDIAF$_{447}$ |
| GalNAc-T2-t3$_{Flexible\ linker}$ (**Chimera 1**) | P$_{494}$EVYVPDLNPVIS$_{506}$ |
| GalNAc-T2-t3$_{Flexible\ linker-AF}$ (**Chimera 2**) | P$_{494}$EVYVPDLNPVIS$_{506}$AF |
| GalNAc-T2-t4$_{Flexible\ linker}$ (**Chimera 3**) | P$_{433}$NLHVPEDRPGWH$_{445}$ |
| GalNAc-T2-t3$_{Flexible\ linker-AF-P503A}$ (**Chimera 2_P503A**) | P$_{494}$EVYVPDLNAVIS$_{506}$AF |
| GalNAc-T2 (**double mutant**, **R438A-D444A**) | P$_{435}$EL**A**VPDHQ**A**IAF$_{447}$ |
| GalNAc-T2 (**triple mutant**, **R438A-D444A-F447A**) | P$_{435}$EL**A**VPDHQ**A**IA**A**$_{447}$ |
| GalNAc-T3 | P$_{494}$EVYVPDLNPVIS$_{506}$ |
| GalNAc-T4 | P$_{433}$NLHVPEDRPGWH$_{445}$ |
| GalNAc-T4-t2$_{Flexible\ linker}$ (**Chimera 4**) | P$_{435}$ELRVPDHQDI$_{445}$ |
| GalNAc-T4-t2$_{Flexible\ linker-AF}$ (**Chimera 5**) | P$_{435}$ELRVPDHQDIAF$_{447}$ |
| GalNAc-T4-t2$_{Flexible\ linker-A}$ (**Chimera 6**) | P$_{435}$ELRVPDHQDIA$_{446}$H |

Above are the sequences of the native transferases and linker domain mutants and chimeras utilized in this work. Only the sequences of the linkers that are exchanged are shown. Shortened names in (bold) are referred to in the text and figures

[a]Note that some of the exchanged linker regions also include part of the succeeding β-strand of GalNAc-T2 and GalNAc-T4. This is due to the boundaries of flexible linkers are not well defined at the C terminus. The additional Pro in the flexible linker of GalNAc-T3/T4, that is not present in GalNAc-T2, is underlined

interactions in GalNAc-T2, which do take in place in GalNAc-T4 (see for instance the persistent salt bridge between Arg397 and Glu487 in Supplementary Movie 5). Thus, the intrinsic conformational preference(s) of a short, unfolded fragment of ~13 residues, was found to determine the relative orientation of two very large protein domains (see also Supplementary Movies 6 and 7; these movies show the intrinsic different conformational preferences of the GalNAc-T2 (folded) and GalNAc-T3 (extended)

flexible linkers by performing 500 ns MD simulations of these isolated fragments).

**Kinetics characterization of the chimeras.** To validate our MD simulations predictions, we expressed and purified **chimeras 1–3** and performed a series of partial and full kinetic analysis against our peptide substrates (Fig. 4b, c) (also note that **chimeras 4–6**

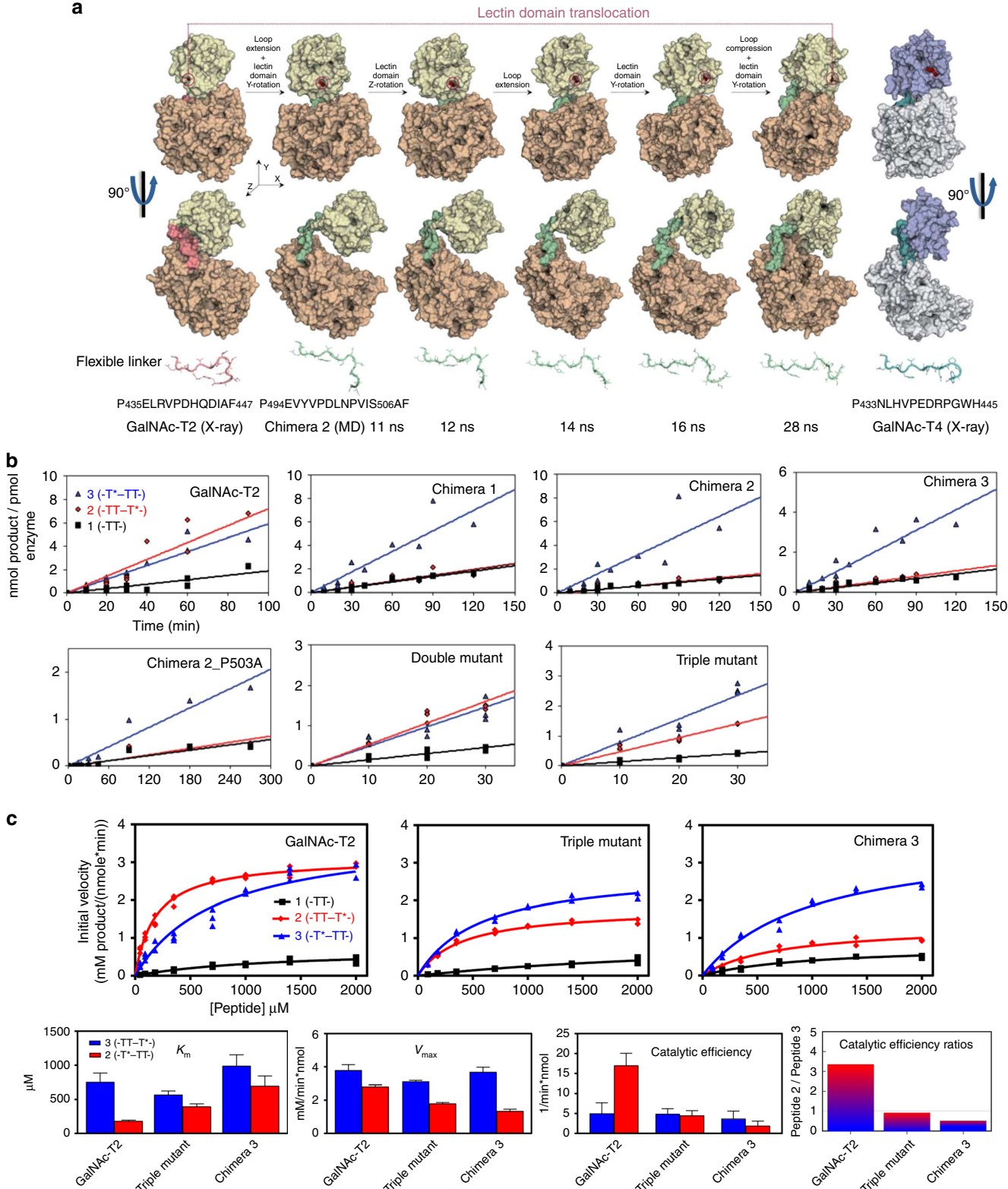

could not be expressed in our hands, see "Methods" section and Table 2). Remarkably, the GalNAc-T2 chimeras rendered glycosylation preferences similar to GalNAc-T3 and GalNAc-T4[15, 16] (preferring monoglycopeptide **3** over monoglycopeptide **2**, by approximately fourfold, which is significantly different from native GalNAc-T2) despite comprising most of the GalNAc-T2 architecture (Fig. 4b, c and Supplementary Fig. 9). Interestingly, GalNAc-T2's preference for these substrates varied with peptide substrate concentration, ranging from a high preference for monoglycopeptide **2** at low substrate concentrations to nearly equal preferences for both monoglycopeptides **2** and **3** at higher substrate concentrations (Fig. 4c). These differences arise from monoglycopeptide **2**'s ~fourfold lower $K_m$ compared to monoglycopeptide **3**, thus giving an over threefold increase in catalytic efficiency for monoglycopeptide **2** over monoglycopeptide **3** (Fig. 4c and Supplementary Table 2). These findings readily explain the previously reported differences in GalNAc-T2's long-range glycosylation preferences[15, 17]. This type of behavior was not observed in the kinetics of GalNAc-T4 or **chimera 3** (note that detailed kinetics were not performed on **chimera 1** or **2**). Further analysis revealed that the changes in specificity for **chimeras 1–3** were chiefly due to a decrease in the observed specific activity (**chimera 1** and **2**), and changes in $V_{max}$ and $K_m$ (**chimera 3**) against monoglycopeptide **2** compared to that of monoglycopeptide **3** whose specific activity and kinetic parameters remained relative constant (Fig. 4b, c, Supplementary Table 2, and Supplementary Fig. 9). It is also worth noting that the specific activities (and detailed kinetics when obtained) for **chimeras 1–3** toward the "naked" peptide **1** are quite similar to those of GalNAc-T2, indicating that the GalNAc-T3/T4 flexible linkers in these chimeras did not affect the overall architecture of the GalNAc-T2's active site (Fig. 4b, c, Supplementary Table 2, and Supplementary Fig. 9). Finally, we evaluated which Thr residues of the substrates were preferably glycosylated by Edman amino acid sequencing, finding that the Thr residue adjacent to the PXP motif was glycosylated by **chimeras 1–3**, as would be expected for the PXP motif[16] (Supplementary Fig. 10). Little to no glycosylation was observed at the proximal Thr.

Together, our results suggest that the glycosylation preferences found for the GalNAc-T2 chimeras can be reasonably explained by the rotation of the lectin domain as predicted by the MD simulations of **chimera 2**. Thus, the lectin domains of these chimeras must begin to adopt an equivalent position to that of GalNAc-T4 to account for their altered long-range glycosylation preferences. We conclude that the flexible linker modulates the rotation of the lectin domain, which subsequently determines the orientation of the functional GalNAc-binding site of the lectin domain with respect to the catalytic domain. This modulation of the lectin domain relative to the catalytic domain likely determines the distinct long-range glycosylation preferences for all of the GalNAc-T isoforms. Our studies below further examine the role of the linker in dictating these long-range preferences.

**Site-directed mutagenesis of the GalNAc-T2 and chimera 2.** Next, we explored whether specific particular residues within the flexible linkers might be responsible for the lectin domain orientation. A multiple alignment of the flexible linkers (Supplementary Fig. 7) shows that GalNAc-T3/T4/T6/T12, which share the same long-distance glycosylation preferences[15, 16], contain an additional Pro that is not present in GalNAc-Ts encompassing the opposed glycosylation preferences such as GalNAc-T1/T2/T14. This Pro in the flexible linkers of GalNAc-T3 (**P**$_{494}$EVYV**P**DLN**P**VIS$_{506}$; underlined) and GalNAc-T4 (**P**$_{433}$NLHV**P**EDR**P**GWH$_{445}$; underlined) is located at the end of the loop with respect to the flexible linker of GalNAc-T2 (**P**$_{435}$ELRV**P**DHQDIAF$_{447}$). Considering that Pro residues are abundant in protein turns, we hypothesized that this additional Pro might be, at least in part, responsible of the different orientation found crystallographically for the GalNAc-T4 lectin domain, and computationally for **chimera 2**. To explore this hypothesis, we mutated this Pro (in the GalNAc-T3 flexible linker sequence of the GalNAc-T2 **chimera 2**) to Ala-producing **chimera 2_P503A** with the expectation that it would reverse its preference (Table 2). This chimera, however, showed an identical glycosylation preference as the starting **chimera 2** (Fig. 4b and Supplementary Fig 9), rather than reverting to the expected wild-type GalNA-T2 preference. This suggests that more complex events within the flexible linker might take place to explain the interplay between the flexible linker dynamics and its coupling with the lectin domain rotation. Also note that this mutated chimera displays a very low-specific activity compared to GalNAc-T2 (~3–10%) (Supplementary Fig. 9). This may suggest that the architecture of the catalytic site of this chimera may have been altered or perhaps blocked by an altered orientation of its lectin domain.

Next, we evaluated in detail the interactions between residues of the flexible linker in the compact and extended crystal structures of GalNAc-T2[4]. Two major interactions, namely a salt bridge between Arg438 and Asp444, and a CH–π interaction between Pro440 and Phe447, appeared to be important to maintain the compact, folded structure of its flexible linker (Supplementary Fig. 11). We reasoned that these interactions might be important for fixing the GalNAc-T2 lectin domain orientation with respect to its catalytic domain. To test this hypothesis, we generated the double mutant R438A-D444A and the triple mutant R438A-D444A-F447A (Table 2). Our assumption was that the disruption of these interactions might lead to more "flexible" linkers, which in turn could cause some degree of rotation of the lectin domain toward the orientation observed in GalNAc-T4. MD simulations showed that, whereas the lectin domain did not rotate in the appropriate direction in the double mutant R438A-D444A (Supplementary Movie 8), it did undergo complete rotation in the triple mutant R438A-D444A-F447A toward the orientation observed in **chimera 2** and GalNAc-T4, although such motion required a much longer simulation time (complete rotation observed after 160–450 ns, Supplementary Movie 9).

**Fig. 4** Characterization of the GalNAc-T2 chimeras and mutants. **a** MD simulations in explicit water of **chimera 2** (500 ns total simulation time; water molecules were removed for clarity). The 28 ns time-lapse snapshots show the dynamic events occurring during the lectin domain reorientation observed for this chimera. The GalNAc-T2 (far left) and GalNAc-T4 (far right) crystal structures are shown as references for the initial and final states. All structures are illustrated in a surface view with two different orientations. The lectin and catalytic domains, and the flexible linker of GalNAc-T2 and GalNAc-T4 are depicted as yellow/purple blue, orange/light gray, and red/deep teal, respectively. The colors used for the **chimera 2** and GalNAc-T2 are the same except for the flexible linker of the former, which is pale green. Flexible linkers are also shown at the bottom in a cartoon- and sticks-like view. **b** Glycosylation time course plots of GalNAc-T2 and the GalNAc-T2 chimeras and mutants against (glyco)peptides **1–3** at substrate concentration of 1.4 mM. The obtained specific activities and selected substrate activity ratios are given in Supplementary Fig. 9. **c** Complete glycosylation kinetics (initial specific activity versus substrate concentration) for GalNAc-T2, the GalNAc-T2-triple mutant and the GalNAc-T2 **chimera 3** against substrate (glyco)peptides **1–3** (top panels) with plots of the obtained Michaelis–Menten kinetic parameters, $K_m$, $V_{max}$, catalytic efficiency ($V_{max}/K_m$), and catalytic efficiency ratios (monoglycopeptide **2** over **3**) (bottom panels). Kinetic parameter values are summarized in Supplementary Table 2

These results were validated by their observed substrate preferences (obtained for both mutants; Fig. 4b and Supplementary Fig. 9) and by a kinetic analysis of the triple mutant R438A-D444A-F447A (Fig. 4c and Supplementary Table 2). Whereas the double mutant gave nearly equal monoglycopeptide preferences as observed for GalNAc-T2 (at high substrate concentration), the triple mutant showed a 1.5–2-fold preference for monoglycopeptide **3** over monoglycopeptide **2**, resembling the preference of GalNAc-T4 (Fig. 4b,c, Supplementary Table 2, and Supplementary Fig. 9). As with **chimeras 1–3**, the change in preferences observed with the GalNAc-T2 triple mutant was due to a decrease in $V_{max}$ and increase in $K_m$ for monoglycopeptide **2** rather than due to a significant change in the kinetic properties of monogycopeptide **3** (Fig. 4b,c, Supplementary Table 2, and Supplementary Fig. 9). The specific activity (and kinetic constants, when obtained) of these linker mutants against peptide **1** are very similar to GalNAc-T2, as well as **chimeras 1–3**, further supporting the notion that their catalytic domains have not been significantly altered (Supplementary Table 2, Fig. 4b, c, and Supplementary Fig. 9). In summary, the stepwise change in preference (Fig. 4b, c) from monoglycopeptide **2** toward that of monoglycopeptide **3**, for native GalNAc-T2, the triple mutant and **chimera 3**, mostly reflects changes in the kinetic constants against monoglycopeptide **2** rather than those for monoglycopeptide **3**.

All together, these results strongly suggest that the flexible linker plays a significant role in directing the long-range glycopeptide specificity of this family of transferases by altering the relative orientations of the catalytic and lectin domains. In addition, our results suggest that specific interactions within the flexible linker are sufficient to modulate this orientation thereby dictating each isoforms long-range glycosylation preferences.

## Discussion

Most types of protein O-glycosylation in higher eukaryotes (O-Man, O-Xyl, O-Glc, O-GlcNAc, and O-Fuc) are initiated by only one or two GT isoenzymes[1]. However, these types of protein O-glycosylation are only found on limited proteins and sequence motifs. On the contrary, Nature has imposed a large family of GalNAc-Ts isoforms to account for mucin-type O-glycosylation. Initially, these GTs were considered redundant isoenzymes but recent findings clearly demonstrate that individual GalNAc-T isoforms serve unique biological functions in health and disease. For example, deficiency in *GALNT3* and *GALNT*2 causes familial tumoral calcinosis[24] and dyslipidemia[25], respectively. How such transferase specificity may be regulated is best illustrated by GalNAc-T3, which is the only isoform capable of glycosylating Thr178 at a proprotein convertase (PC) processing site (RHTR[179]↓) of FGF23, an important regulator of phosphate homeostasis[22]. However, glycosylation of Thr178 by GalNAc-T3 requires prior O-glycosylation of residue Thr171 of FGF23, demonstrating that GalNAc-T3 utilizes its long-range N-terminal preference for prior GalNAc glycosylation to target or amplify the glycosylation of Thr178[15]. Thus, the combination of local peptide sequence and remote prior glycosylation is used by these transferases to ensure the fidelity of isoform-specific glycosylation. With the recent quantitative differential O-glycoproteomics studies[7], it is now clear that many individual GalNAc-T isoforms serve unique non-redundant contributions to glycosylation of the proteome, presumably employing similar strategies as that of GalNAc-T3 and FGF23. The long-range lectin domain-mediated functions of GalNAc-Ts are likely instrumental for the coordinated glycosylation of additional diverse sequence motifs and in particularly high-density acceptor sequence motifs such as in mucins. Therefore, it is likely that the large number of GalNAc-Ts

and their lectin domains evolved to recognize diverse peptide sequence motifs as well as to differentially recognize prior sites of GalNAc glycosylation. This would ensure the fidelity of glycosylation in both low- and high-density glycosylated regions by sequential orchestrated glycosylation reactions, coordinated by both the specificity of the catalytic domain as well as the long-range recognition of the lectin domain, that interacts with the prior attached GalNAc residue[15, 16]. Therefore, to fully understand the biological functions of these enzymes and to elucidate their roles in disease, we must understand the molecular basis, and mechanisms therein, which dictate and modulate their complex long- and short-range substrate specificity.

Here we have addressed the molecular mechanisms behind the unique long-range lectin domain-dependent glycosylation functions of GalNAc-Ts and have provided a model explaining how these transferases differentially recognize such N- or C-terminal long-range prior glycosylation[15, 16]. We previously reported that the location of the lectin α-subdomain GalNAc-binding site, relative to the catalytic domain, was key to explaining how GalNAc-T2 delivered N-terminal acceptor sites of C-terminal monoglycopeptides to the catalytic domain in a lectin domain-dependent manner[4]. We also demonstrated that the flexible linker, located ~28–36 Å from the active site of GalNAc-T2, and ~18–25 Å from GalNAc-T2's lectin α-subdomain was in charge of the conformational heterogeneity of GalNAc-T2[4]. This was clearly due to the translational capacity of the lectin domain with respect to the catalytic domain, which was further modulated in the presence of substrates[4]. Note also that in all of the GalNAc-T2 structures reported, irrespectively of whether they are compact or extended structures, the lectin domain is positioned such that its α-subdomain GalNAc-binding site is always located on the left side of the lectin domain[4, 11, 12]. However, it has remained a conundrum how other GalNAc-Ts (i.e., GalNAc-T3, T4, T12, etc.) presented the opposite long-range glycosylation preference of GalNAc-T2 and others. Our multidisciplinary approach presented herein has revealed additional conformational properties of the short flexible linker that provide the molecular basis for the distinct long-range glycosylation preferences found for these enzymes. Our crystal structure of GalNAc-T4 revealed that its lectin domain is positioned such that its lectin α-subdomain GalNAc-binding site is now located on the right side of the lectin domain, in agreement with our proposed Model 1 (Figs. 1c and 3). This is fundamental to explaining why GalNAc-T4 and similar behaving GalNAc-Ts prefer to glycosylate C-terminal acceptor sites of N-terminal glycosylated glycopeptides. Our STD-NMR studies together with the transferase kinetic studies suggested that the orientation of the lectin domain-bound GalNAc moiety was fundamentally important for the correct orientation and presentation of the acceptor residues to the catalytic site, thereby leading to more efficient catalysis. Our MD simulations on multiple chimera transferases together with their kinetic analysis, further provide compelling evidence that the flexible linkers not only allow translational motion to the lectin domain, but also rotational motion. Both types of motions likely differ between the GalNAc-T isoforms and therefore will contribute to the different glycosylation preferences found among these enzymes. All together, our studies point to the dissimilar flexible linkers as major contributors to the distinct long-range glycosylation preferences of these transferases (Supplementary Fig. 7). It is further likely that some flexible linkers will have conformational properties in solution that allow two or more conformational states of the lectin domain relative to the catalytic domain thus accounting for the dual long-range preferences observed for GalNAc-T2 (and its chimeras and mutants) as well as the several GalNAc-Ts that display nearly equal N- and C-terminal long-range glycosylation preferences (i.e., GalNAc-T5, 13,16[15]). That we did not observe

such conformational flexibility in our MD trajectories of GalNAc-T2 (and some chimeras and mutants) may suggest that the activation energy for this interconversion may be very high and/or that our simulations were not sufficiently long enough.

In conclusion, we have provided for the first time the molecular basis for the distinctive long-range glycosylation preferences of the GalNAc-Ts, which is based on a very small flexible linker that provides rotational and translational capacity to the lectin domain. This work further exemplifies how a structural feature, very distant from both the active site and the lectin domain GalNAc-binding site, is capable of tuning the activity and specificity of these biologically important multidomain enzymes.

## Methods

**Cloning and purification of human GalNAc-T4/chimeras**. The DNA sequence encoding amino acid residues 36–578 of the human GalNAc-T4, defined as *galnact4*, was codon optimized and synthesized by GenScript (USA) for expression in *Pichia pastoris*. The DNA, containing at the 5′ end a recognition sequence for *Xho*I and a KEX2 cleavage signal, and at the 3′ end a sequence encoding a histidine tag, a stop codon and a recognition sequence for *Sac*II, was cloned into the pUC57 vector (GenScript). Following digestion with *Xho*I and *Sac*II, the construct was subcloned into the protein expression vector pPICZαA, resulting in the expression plasmid pPICZαA-*galnact4*. Subsequently, the plasmid was linearized by *Sac*I-HF and transformed into SMD1168. Transformants were selected and colonies were grown as described before[26, 27]. The DNA sequences encoding the GalNAc-T4 chimeras were also synthesized by GenScript and cloned in the same above vector. DNA linearization and transformation in SMD1168 were also performed in the same manner described above. Note that none of the GalNAc-T4 chimeras were expressed as a soluble secreted form in *P. pastoris* supernatants, impeding their biophysical characterization.

Supernatant containing the human GalNAc-T4 was dialyzed against buffer A (20 mM $Na_2HPO_4$ pH 7.4, 20 mM imidazol 500 mM NaCl) and loaded into a His-Trap Column (GE Healthcare). The protein was eluted with an imidazol gradient in buffer A from 20 mM to 500 mM. Buffer exchange of GalNAc-T4 into buffer B (25 mM Tris pH 8, 150 mM NaCl) was carried out using a HiPrep 26/10 desalting column (GE Healthcare). GalNAc-T4 was further purified by size exclusion chromatography using a Superdex 75 XK26/60 column (Sigma) previously equilibrated with buffer B. Fractions containing GalNAc-T4 were dialyzed against buffer C (25 mM Tris pH 8, 1 mM TCEP), concentrated and used for biophysical experiments.

**Cloning and purification of GalNAc-T2 chimeras/mutants**. The DNA sequences encoding the GalNAc-T2 chimeras were synthesized by GenScript and cloned in the vector pPICZαA vector as previously described[12]. Site-directed mutagenesis experiments were also performed by GenScript using the previous reported template pPICZαA*galnact2* (K75-Q571)[12] and the plasmid encoding the **chimera 2**. The linearization and transformation of all constructs to SMD1168 were also performed in the same manner described for GalNAc-T4. The purification of all these GalNAc-T2 chimeras and mutants were expressed and purified using the purification protocol of the wild-type enzyme, as described previously[12]. Purity of enzymes was evaluated by SDS-PAGE coomassie staining and quantification of enzymes was quantified by absorbance at 280 nm using their theoretical extinction coefficients.

**Crystallization**. Crystals of the GalNAc-T4 were grown by hanging drop experiments at 18 °C by mixing 0.5 μl of protein solution (4 mg/ml GalNAc-T4, 5 mM UDP, 2 mM $MnCl_2$, and 5 mM MUC5AC-3,13[4] (GTT*PSPVPTTSTT*SAP) in 25 mM Tris pH 7.5, 0.5 mM EDTA, and 1 mM tris(2-carboxyethyl)phosphine (TCEP)) with an equal volume of a reservoir solution (18% PEG3350, 0.1 M ammonium nitrate). Under these conditions, crystals appeared within 2–5 days. Note that MUC5AC-3-13 acted as an additive to improve the diffraction quality and the resolution of the crystals. The crystals were soaked for 30 min with a mix containing 20 mM monoglycopeptide **3** and 20 mM UDP in 25 mM Tris pH 7.5 and 2 mM $MnCl_2$. Then, the crystals were cryo-protected with 25% ethylene glycol, 18% PEG3350, and 0.1 M ammonium nitrate, and frozen in a nitrogen gas stream cooled to 100 K.

**Structure determination and refinement**. The data were collected in the beamline I03 of Diamond (DLS) at a wavelength and temperature of 0.97 Å and a temperature of 100 K, respectively. The data were processed and scaled using the XDS package[28] and CCP4[29, 30] software. Relevant statistics are given in Supplementary Table 1. The crystal structure was solved by molecular replacement with Phaser[29, 30] and using the PDB entry 5AJP as the template that corresponds to the human GalNAc-T2. Initial phases were further improved by cycles of manual model building in Coot[31] and refinement with REFMAC5[32]. Once the GalNAc-T4 catalytic domain was unambiguously built and refined, ARP/wARP[29, 30] was used

to fully build the GalNAc-T4 lectin domain. Again, new rounds of Coot and refinement with REFMAC5 were performed. The crystal structure of GalNAc-T4 from crystals co-crystallized with MUC5AC-3-13 displayed no electron density for the MUC5AC-3-13 diglycopeptide. The final model of crystals soaked with monoglycopeptide **3** and UDP was validated with PROCHECK, model statistics are given in Supplementary Table 1. The asymmetric unit of the triclinic crystal contained two molecules of GalNAc-T4. Despite the long soaking and high concentration of monoglycopeptide **3** and UDP, only clear density was visualized for the presence of GalNAc-O-Thr and GalNAc moieties of monoglycopeptide **3** bound to the lectin α-subdomain GalNAc-binding site (Fig. 3a, c). The Ramachandran plot shows that 96.71, 2.89, and 0.40% of the amino acids are in most favored, allowed, and disallowed regions, respectively.

**Surface plasmon resonance experiments**. SPR experiments were performed at 25 °C with a Biacore X-100 apparatus (Biacore AB) in 25 mM Tris buffer, 1 mM DTT, 4 mM $MnCl_2$, 100 μM UDP, 0.01% surfactant P20, pH 7.5 (running buffer) at 25 °C. Flow cells (CM5 sensor chip; Biacore) were activated for 7 min by injecting 140 μl of 50 mM N-hydroxysuccinimide (NHS): 200 mM ethyl-3(3dimethylamino) propylcarbodiimide (EDC). GalNAc-T4 was immobilized on a flow cell 2 by injection of a 100 μg/ml protein solution diluted with 10 mM sodium acetate buffer with a flow rate of 10 μl/min for 7 min followed by an injection of 130 μl ethanolamine to block any remaining activated groups on the surface. The level of immobilization reached was about 8000 RU. Flow cell 1, used as reference, was blocked with ethanolamine at the same conditions of flow cell 2 without immobilization of protein. Affinity experiments were made using a series of different concentrations of peptide **3** in the range of 0.01–5 mM with a flow rate of 30 μl/min during 100 s. Each injection was followed by a 100 s injection of running buffer (dissociation phase). No regeneration steps were performed between injections. Response data were collected at real time and analyzed with the Biacore® X-100 Evaluation software and plotted as response shift versus analyte concentration. We could not determine a $K_d$ value due to binding saturation was not achieved.

**Synthesis of peptides**. Peptides were synthesized via stepwise microwave-assisted solid-phase peptide synthesis on a Liberty Blue synthesizer using the Fmoc strategy on Rink Amide MBHA resin (0.1 mmol). The glycosylated amino acid building blocks (2.0 equiv) were synthesized as described in the literature[33] and manually coupled using HBTU, while the other Fmoc amino acids (5.0 equiv) were automatically coupled using oxyma pure/DIC. The *O*-acetyl groups of $(AcO)_3GalNAc$ moiety were removed in a mixture of $NH_2NH_2$/MeOH (7:3). The peptides were then released from the resin, and all acid sensitive side-chain protecting groups were simultaneously removed using TFA 95%, TIS 2.5%, $H_2O$ 2.5%, followed by precipitation with cold diethyl ether. Finally, they were purified by HPLC using a Phenomenex Luna C18(2) column (10 μ, 250 mm × 21.2 mm) and a dual absorbance detector, with a flow rate of 20 ml/min. Peptides were further subjected to Edman amino acid sequencing on a Shimadzu PPSQ-53A peptide sequencer prior to use.

Peptide **1**: HPLC: Rt = 11.53 min (Grad: water 0.1% TFA/acetonitrile (95:5) → (87:13), 13 min, λ = 212 nm). HRMS ESI + (*m/z*) calcd. for $C_{39}H_{66}N_{14}O_{15}$ [M + 2 H]$^{2+}$ 485.2411, found 485.2301.

Monoglycopeptide **2**: HPLC: Rt = 11.75 min (Grad: water 0.1% TFA/acetonitrile (95:5) → (85.6:14.4), 15 min, λ = 212 nm). HRMS ESI + (*m/z*) calcd. for $C_{57}H_{95}N_{18}O_{24}$ [M + H]$^+$ 1415.6761, found 1415.6739.

Monoglycopeptide **3**: HPLC: Rt = 13.72 min (Grad: water 0.1% TFA/acetonitrile (93:7) → (86:14), 15 min, λ = 212 nm). HRMS ESI + (*m/z*) calcd. for $C_{56}H_{93}N_{18}O_{24}$ [M + H]$^+$ 1401.6605, found 1401.6546.

**NMR experiments**. All NMR experiments were recorded on a Bruker Avance 600 MHz spectrometer equipped with a triple channel cryoprobe head. The$^1$H NMR resonances of the peptides **1–3** were completely assigned through standard 2D-TOCSY (30 and 80 ms mixing time) and 2D-NOESY experiments (400 ms mixing time). Solution conditions used for the NMR characterization studies were 1–3 mM (glyco)peptide, 25 mM perdeuterated tris-d11 in 90:10 $H_2O$/$D_2O$, 7.5 mM NaCl, and 1 mM DTT, uncorrected pH 7.4. The assignments were accomplished either at 278 or 298 K. The resonance of 2,2,3,3-tetradeutero-3-trimethylsilylpropionic acid (TSP) was used as a chemical shift reference in the$^1$H NMR experiments (δ TSP = 0 ppm). Peak lists for the 2D-TOCSY and 2D-NOESY spectra were generated by interactive peak picking using the computer aided resonance assignment (CARA) software.

Samples for STD experiments were prepared in perdeuterated 25 mM TRIS-d11 in deuterated water, 7.5 mM NaCl, and 1 mM DTT, uncorrected pH 7.4. STD-NMR experiments were performed at 298 K in the presence of 75 μM UDP, 75 μM $MnCl_2$ with ~880 μM peptide (or GalNAc-O-Me) and 13.5 μM GalNAc-T4 giving a molar ratio of 65:1 peptide:GalNAc-T4.

The STD-NMR spectra were acquired with 1920 transients in a matrix with 64 k data points in t2 in a spectral window of 12335.53 Hz centered at 2819.65 Hz. An excitation sculpting module with gradients was employed to suppress the water proton signals. Selective saturation of the protein resonances (on resonance spectrum) was performed by irradiating at −1 ppm using a series of Eburp2.1000-shaped 90° pulses (50 ms, 1 ms delay between pulses) for a total saturation time of

2.0 s. For the reference spectrum (off resonance), the samples were irradiated at 100 ppm. Proper control experiments were performed with the ligands in the presence and absence of the protein in order to optimize the frequency for protein saturation (−1 ppm) and to ensure that the ligand signals were not affected. However, all the glycopeptides when irradiated at −1 ppm in the absence of protein showed residual saturation on the aliphatic methyl groups in the STD-NMR spectra. This nonspecific saturation was taken into account, by subtraction, when quantifying the STD-NMR data in the presence of the transferase. As well, a blank STD experiment with only the protein was also recorded. The substraction of this protein STD spectrum allowed eliminating the signal background of the protein. In all cases, to accomplish the STD-NMR-derived epitope mapping of each ligand, the STD-NMR total intensities were normalized with respect to the highest STD-NMR response. For Fig. 2b and Supplementary Fig. 1, the STD response of each amino acid corresponds to the average of STD percentages of all amino acid proton resonances that were measured with sufficient accuracy. The signal of the anomeric proton of as well as, the Hα protons of the Ala amino acids of glycopeptides could not be analyzed in the STD-NMR spectra due to their close proximity to the HDO resonance. Proton resonances, from Gly and Pro, appear in the same chemical shift region of the spectrum and were not discriminated.

**Molecular docking**. Two different docking calculations were conducted to generate the ternary complex.

First, UDP was docked into GalNAc-T4 with the aid of of AutoDock Vina 1.1.2[34]. The predicted binding energies ranged from −6.4 to −5.2 kcal/mol. The Autogrid grid point spacing was set at 0.375 Å, center coordinates of the grid box were 5.4, −2.4, − 7.2 ($x$, $y$, $z$), and number of grid points in $xyz$ was 76, 40, 42, respectively. All allowed torsional bonds were considered rotatable. The 3D structure of the docked ligand to the protein with the lowest-binding energy was used for further calculations (see below).

In a second simulation, glycopeptide T(α-$O$-GalNAc)GAGAGAGTTPGPG was docked into GalNAc-T4. The Autogrid grid point spacing was set at 0.375 Å, center coordinates of the grid box were 0.0, −9.7, 9.4 ($x$, $y$, $z$), and number of grid points in $xyz$ was 86, 60, 90, respectively. All allowed torsional bonds were considered rotatable, except the bonds involved in the glycosidic linkage, which were fixed to $\phi = 64°$, $\psi = 110°$. The 3D structure of the docked ligand to the protein with lowest-binding energy and with the GalNAc unit located in the lectin domain region, was then used for further calculations.

The combination of both structures was used to build up the ternary complex, which was subjected to MD simulations (see below).

**MD simulations**. The starting coordinates for the complex between glycopeptide 3, UDP/Mn$^{+2}$, and GalNAc-T4 were generated combining the X-ray structure of GalNAc-T4 solved in this manuscript and the crystal structure of the activated form of GalNAc-T2 in complex with UDP/Mn$^{+2}$ and MUC5AC-13 (PDB entry 5AJP). The GalNAc unit of the glycopeptide 3 was located by superimposition to the one solved in the X-ray structure of GalNAc-T4, and UDP was located by superimposition of GalNAc-T2 and T4 proteins. The peptide backbone of compound 3 was situated so that the Thr residues were laying on the proximity of the UDP moiety.

The starting coordinates for **chimera 2** were generated by superimposing the X-ray structures of GalNAc-T2 (PDB entry: 5AJP) and GalNAc-T4. Residues 435–445 (PELRVPDHQDI) were manually deleted from GalNAc-T2 and replaced by residues 494–506 (PEVYVPDLNPVIS) of GalNAc-T3 using PyMol (http://www.pymol.org). Residues $A_{507}F_{508}$ were maintained in GalNAc-T2 to facilitate the linkage between its catalytic and lectin domains. The double (R438A-D444A) and triple mutant (R438A-D444A-F447A) were generated using PyMol. The same protocol was used for other computational chimeras.

Force field parameters for the substrates were generated with the antechamber module of Amber14 using a combination of GLYCAM06[35] parameters for the GalNAc unit and the general Amber force field (GAFF) for GDP, with partial charges set to fit the electrostatic potential generated with HF/6-31 G(d) by RESP. The charges are calculated according to the Merz–Singh–Kollman scheme using Gaussian 09[36]. Each protein was immersed in a truncated octahedral box with a 10 Å buffer of TIP3P water molecules and neutralized by adding explicit counter ions (Na$^+$, Cl$^-$). All subsequent simulations were performed using the $ff14SB$ force field[37]. A two-stage geometry optimization approach was used. The first stage minimizes only the positions of solvent molecules and ions, and the second stage is an unrestrained minimization of all the atoms in the simulation cell. The systems were then gently heated by incrementing the temperature from 0 to 300 K under a constant pressure of 1 atm and periodic boundary conditions. Harmonic restraints of 30 kcal/mol were applied to the solute, and the Andersen temperature coupling scheme was used to control and equalize the temperature. The time step was kept at 1 fs during the heating stages. Water molecules are treated with the SHAKE algorithm such that the angle between the hydrogen atoms is kept fixed. Long-range electrostatic effects are modeled using the particle-mesh-Ewald method. An 8-Å cutoff was applied to Lennard–Jones and electrostatic interactions. Each system was equilibrated for 2 ns with a 2-fs time step at a constant volume and temperature of 300 K. Production trajectories were then run for additional 100–500 ns under the same simulation conditions.

**Transferase assays and kinetics**. Specific activity determinations: GalNAc-T glycosylation reactions against (glyco)peptides 1–3 were performed with 75 mM sodium cacodylate, pH 6.5, 1 mM 2-mercaptoethanol, 10 mM MnCl$_2$, 0.25 mM [$^3$H]-radiolabeled UDP-GalNAc (~6 × 10$^8$ DPM/μmole, American Radiolabeled Chemicals Inc.), and 1.4 mM (~1.25 mg/ml) of (glyco)peptide (from 4 mM stock in 1 mM TRIS, pH ~7) and varying concentrations (0.01 to 0.5 μM) of transferase (determined by OD$_{280}$), giving a final reaction volumes of 20 μl in 100 μl capped Eppendorf tubes. Reactions were incubated at 37 °C in a thermostated microplate shaker (Taitec Microincubator M-36) and quenched at the appropriate reaction time by the addition of 20 μl of 250 mM EDTA and placed on ice prior to final workup. Typically, for each transferase, glycosylation reactions were performed at the same time with all three (glyco)peptide substrates (including a no peptide control) using the same batch of transferase and the same UDP-[$^3$H]-GalNAc stock. Before performing time course experiments, the relative activity of each transferase was determined by trial and error against all three peptides to determine the optimal transferase concentration or incubation time that gave no more than 10% peptide glycosylation against the best substrate. Final time course experiments typically consisted of incubation times of 10, 20, and 30, or 30, 60, and 90 or 120 min (depending on activity) and were performed at least twice. Most transferases were also characterized at two different enzyme concentrations (0.01–0.03 and 0.2–0.5 μM) and the results combined. After quenching the reaction and diluting to 4 ml, free UDP-GalNAc and non-hydrolyzed UDP-GalNAc were removed by passage over a Dowex 1 × 8 anion exchange resin (~3 ml column). Total UDP-[$^3$H]-GalNAc utilization (transfer to peptide substrate and transfer to water, i.e, hydrolysis) was determined by difference after scintillation counting (Beckman LS5801 scintillation counter) 1/20 of the sample before and after passage over Dowex 1 × 8. The extent of [$^3$H]-GalNAc transfer to peptide and the extent of hydrolysis was determined by Sephadex G10 gel filtration analysis, typically for the longest reaction time point, as previously described[15]. Example gel filtration chromatograms are given in Supplementary Fig. 12. Significant hydrolysis was only observed for GalNAc-T4, which was corrected for in determining its specific activity. Specific activity was obtained by calculating at each individual data time point (of the combined time course plots, Fig. 4b), an individual-specific activity, which was then averaged to obtain the overall average specific activity (and standard deviation). Specific activity is reported as nmol of glycopeptide product formed per min, per pmol of enzyme. Substrate activity ratios (i.e., substrate ratios 2/1, 3/1, and 3/2) were also obtained at each individual time point and averaged (with standard deviation). These data are tabulated in Supplementary Fig. 9. Note that the Microsoft Excel least squares fitting function LINEST was used to obtain the linear plots shown in Fig. 4b while fixing the $y$ intercept to zero. Note that the specific activity calculated from the least square slope differed slightly (less than 10%) from that determined from the average of the individual points.

Detailed enzyme kinetics on GalNAc-T4, GalNAc-T2, the GalNAc-T2 triple mutant and GalNAc-T2 **chimera 3**: Glycosylation reactions were performed as described above using a fixed enzyme concentration (0.06 μM for GalNAc-T4, and 0.02 μM for GalNAc-T2 and its linker constructs) with varying peptide concentrations (typically 45, 90, 180, 350, 700, 1400, and 2000 μM). Incubation times (10–90 min) were chosen such that no more than 30% of the UDP-GalNAc donor was depleted while typically giving less than 10% peptide glycosylation, after correction for UDP-GalNAc hydrolysis described above. Specific activities were obtained from one to two individual incubation time points performed in two–three separate experiments. These individual-specific activity values were used to calculate the kinetic constants of $K_m$ and $V_{max}$ using the nonlinear Michaelis–Menten fitting program in GraphPad Prism 7.03.

**Determination of site of glycosylation**. The determination of substrate glycosylation sites was performed by Edman amino acid sequencing (Applied Biosystems Procise 494 peptide sequencer) of the G10 isolated [$^3$H]-GalNAc glycosylated substrate as previously described[15, 16] where each cycle was collected on a fraction collector and scintillation counted (Beckman LS5801 scintillation counter) for [$^3$H]-GalNAc content. Although there is commonly sample-to-sample variability in the [$^3$H]-GalNAc content loaded on the sequencer (due sample losses and different initial UDP-[$^3$H]-GalNAc-specific activities), the observed sites of incorporation were found to be identical between different experiments with the same transferase and substrate. The presence of [$^3$H]-GalNAc lag after a peak of [$^3$H]-GalNAc incorporation is commonly observed in these determinations, which is due to the poor extraction from the sample filter of the glycosylated-PTH residues compared with the standard amino acid PTH derivatives[15, 16].

**Data availability**. The coordinate and structure factor for GalNAc-T4 in complex with monoglycopeptide 3 has been deposited in the Worldwide Protein Data Bank (wwPDB) with the accession code 5NQA. All relevant data are available from the authors on reasonable request.

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

## Acknowledgements

We thank synchrotron radiation sources DLS (Oxford) and in particular beamline I03 (experiment number MX10121-7). We thank ARAID, MEC (CTQ2013-44367-C2-2-P, BFU2016-75633-P, CTQ2015-67727-R, CTQ2015-70524-R, and RYC-2013-14706), the National Institutes of Health (GM113534, and instrument grant GM113534-01S), the Danish National Research Foundation (DNRF107), the FCT-Portugal (UID/Multi/04378/2013 and PTNMR Project No 022161), and the DGA (B89) for the financial support. I.C. thanks Universidad de La Rioja for the FPI grant. F.M. thanks FCT-Portugal for IF Investigator. E.L.-N. acknowledges her postdoctoral EMBO fellowship ALTF 1553-2015 co-funded by the European Commission (LTFCOFUND2013, GA-2013-609409) and Marie Curie Actions. H.C. and J.J.-B. thank EU for the TOLLerant project. The research leading to these results has also received funding from the FP7 (2007–2013) under BioStruct-X (grant agreement No. 283570 and BIOSTRUCTX_5186). We also thank BIFI (Memento cluster) and CESGA for computer support.

## Author contributions

R.H.-G. designed the crystallization construct and solved the crystal structure. E.L.-N., M.R., and R.H.-G. purified the enzymes, crystallized the complex, and refined the crystal structure. I.C., F.C., and J.M.P. synthetized the glycopeptides. F.C. and G.J.-O. performed the MD simulations. H.Co., A.D., J.J.-B., and F.M. performed and analyzed the NMR experiments. T.A.G. and E.J.P.D. performed the kinetic studies together with the Edman amino acid sequencing. R.H.-G. wrote the article with the main contribution of T.A.G., F.C., G.J.-O., H.Cl., and F.M. All authors read and approved the final manuscript.

## Additional information

**Competing interests:** The authors declare no competing financial interests.

