## [Peer Review File · Nature Communications]

PEER REVIEW FILE

Reviewers' comments:

Reviewer #1 (Remarks to the Author):

In the present manuscript written by de las Rivas and coworkers, the authors demonstrated to characterize and clarify the function of flexible linker between catalytic and binding domains in acetyl galatosamine transferase (GalNAc-T). Such two-domain enzymes with catalytic and binding domains are quite common structure for degrading and synthesizing enzymes of polysaccharides, while the function, especially interactions between these two domains are less characterized mainly because the linker part between the domains typically flexible, and the domain conformation is thus varied. The authors solved the crystal structure of one of these enzymes with small substrate as a ligand. In addition, they used NMR to determine the substrate conformation combining the molecular dynamic (MD) simulations of intact enzyme to see the conformational change and measured the affinity of synthesized substrates to mutant enzymes. Although each experiment is “typical” approach to clarify each questions, the work combined almost all possible experiments to solve the one important question “how two domains work synergistically” and I was very much enjoyed reading the massive work. Since I guess readers not only in this field but also in general will have same opinion about the work, I would like to suggest publication in Nature Communications after minor correction as follows.

I think the order of the results and balance between the main results and supplementaries are bad. They first show the model of this study, followed by the SPR, structure and MD results. However, the model they showed first looks like summary of this study. Do they need to show the hypothesis first? Considering the variation of the conformation of domains, and the flexibility of substrates, it could be just two but many possibilities can be thought at the beginning of this study. Therefore, If we follow the “logic” of thinking, the model should be much later as a summary. I assume that unbalance situation is happened by the unbalance between main results and supplementaries. Since the data shown in this study are really massive, it should be many variations in the order of figures. Please consider the construction of the manuscript once again.

Is there any sugar moiety in the protein? Using *Pichia pastoris* as a host of recombinant enzyme production, it should have glycosylations. Isn't it affecting to the affinity and structure?

Please add line number for the reviewers convenience. Such considerateness is often important even the content is satisfactory, I think.

Page 3, please add the information of family number for GT and possibly lectin domain.

Page 7, although they used bold font for the compounds, number only is rather difficult to distinguish figure number. Please use "Peptide 1", "Compound 3" or so.

Page 15, "O" should be italic

Page 19, last sentence of first paragraph, How have the authors checked production of enzyme?

Reviewer #2 (Remarks to the Author):

In their manuscript "The interdomain flexible linker of the polypeptide GalNAc transferases (GalNAc-Ts) dictates their long-range glycosylation preferences" de las Rivas et al. investigate the intriguing question of glycopeptide specificity.

While the target has clearly some biomedical potential and is a viable and potentially impactful system to study conclusions about the lectin domain recognition and relative orientation with respect to the catalytic domain remain indirect. While the Xray structure only reveals a structural snapshot and STD NMR shows binding and orientation of the peptide other techniques might prove more convincing. One could possibly study the domain orientations and flexibility of the linker in dependence of the mutants investigated by NMR PREs or single molecule fluorescence/FRET.

With the indirect evidence presented I struggle to see how these results will fundamentally change the field.

Reviewer #3 (Remarks to the Author):

The manuscript by de las Rivas et al, entitled "The inter-domain flexible linker of the polypeptide GalNAc transferases (GalNAc-Ts) dictates their long range glycosylation preferences" involves the use of multiple techniques (X-ray crystallography, NMR, MD simulations, SPR, kinetics assays) to assess long-range GalNAc-glycopeptide binding interactions with the glycosyltransferase GalNAc-T4 that initiates mucin-type O-glycosylation in metazoans. Even though it is already known that GalNAc-T4 and glycosyltransferases in this family demonstrate long-range glycosylation preferences, the mechanism of action is unclear, with a number of models having been presented over the years. This study proposes to validate

one or another of these models. Overall, the work has been well carried out, and the experiments generally have been well performed and are of high quality. However, there are a number of issues that raise concerns.

Experimentally, the best part of the study is the excellent X-ray crystal data and resulting structures. However, the structures and the information from them are not completely novel. E.g. some of the most important interacting residues have already been reported. Thus these structures seem more of a confirmative or supportive nature of what is already known.

The authors used three designed peptides, one un-glycosylated and two O-glycosylated to Thr residues (Table 1) to assess interactions and identify which part of the peptide/Thr is crucial to the interaction. However, their “best” peptide 3 with the N-terminal Thr-GalNAc is similar to that already reported.

SPR was used to quantify peptide binding. Because binding of the peptides is very weak (high mM range), SPR is not the best technique to quantify such extremely weak binding interactions. Why were NMR titrations not performed? In NMR, interactions should be in the fast exchange regime on the chemical shift time scale, and accurate binding constants could likely have been determined.

The STD NMR work provides some evidence for the binding epitope on the glycosylated peptides. However, STD effects are extremely small, with peptide 1 showing no effect, and peptides 2 and 3 showing only very weak effects, with most STD effects being around 1% or less and the maximum effect being around 3.5%. Moreover, STD NMR spectral baselines are not flat and thus likely contribute significantly to error, yet no statistical analyses were performed and values are shown to three significant figures. Therefore, these data are not fully convincing.

MD simulations have been run on the glyco-peptides and enzyme, and they appear to have been technically well done. However, the peptides bound to the enzyme interact in a highly dynamic fashion as stated by the authors (Fig 3e,f). In these instances, simulations have been run for 200 ns, but the stability of the complexes is unclear. No trajectories are shown and no binding energetics are provided. It could be that one needs a longer time frame to assess binding or changes in binding mode, yet no coarse grain simulations have been performed. More MD simulations (500 ns) are shown in Fig S6. However here it may be that rotations do not occur on that time scale. A longer time frame may be needed which coarse grain calculations would provide. Fig 4a also shows structures resulting from MD simulations. Once again, stability of the interactions with the peptides is unclear. For conclusions drawn solely from these MD simulations, experimental validation is needed.

Table 2 shows a list of interesting and potentially informative enzyme linker-chimeras, yet only

three could be expressed. Although limitations on expressing one protein or another is not unusual, it is unfortunate, as the whole set would have been nice to explore. The authors' results here suggest that enzyme chimera glycosylation preferences may be explained by rotation of the lectin domain as seen in MD simulations of chimera 2. However, once again, this is not definitive.

Overall, the findings in this paper are generally interesting, yet mostly suggestive or supportive/confirmatory of what has been already reported. Thus the authors lack concrete proof for any single mechanism or validation of any one model already proposed model. For example, the GalNAc-T4 crystal structure indicates that the GalNAc-binding site of the lectin domain is rotated by 28° relative to the homologous GalNAc-T2 structure. Although the authors state that this provides "an explanation for the long-range glycosylation feature of GalNAc-T4", it in fact only suggests that. And in terms of the flexible linker studies, the authors themselves state that their "flexible linker-swapping constructs of GalNAc-Ts with different glycosylation preferences (only) suggest that the flexible linker dictates rotation of the lectin domain". In some ways, the findings presented here only provide an incremental advance to the field.

Reviewer #4 (Remarks to the Author):

The manuscript presents a structural, (crystallography and simulation) and enzymatic analysis of the GalNAc transferase GalNAc-T4. This transferase has both an enzyme domain and a lectin domain that are combined through a flexible linker. Simulation and enzymatic data are also presented for several chimeras derived by swapping the linkers from GalNAc-T3 and -T4 into GalNAc-T2 (which is closely related in structure to GalNAc-T4). The purpose of the chimeras was to indicate whether changing in linker would switch the substrate glycosylation behavior from normal GalNAc-T2 to that seen in GalNAc-T3/T4. This result was indeed observed.

The results are interpreted as showing that: "the flexible linker causes the rotation of the lectin domain, which determines the orientation of the functional GalNAc-binding site of the lectin domain with respect to the catalytic domain, and in turn leads to the distinct long-range glycosylation preferences of the GalNAc-Ts."

However, the authors also conclude that: " the flexible linkers not only allow translational motion to the lectin domain but also rotational motion".

In my opinion the authors have shown that the differences in glycosylation patterns are "caused" by differences in the relative orientations of the lectin and catalytic domains. But whether or not the linker causes these orientational differences or merely allow them to occur remains unresolved. The authors repeatedly state that the linkers are flexible, as such it is unclear what

might be the driving force that they could exert to cause the changes in domain orientation. In contrast if the linkers were rigid (or at least had strong conformational preferences), it is clear that they could directly control the domain orientations. The data from point mutations provide no clear support for a causative role for the linker.

Minor points that could enhance the manuscript would be:

- 1) inclusion of a graph that plots the domain rotation as a function of the MD simulation time
- 2) Additional clarification of the manner in which the initial structures of the substrate-enzyme complexes were generated. The only information comes from one sentence " The peptide backbone of compound 3 was situated so that the Thr residues were laying on the proximity of the UDP moiety." This description is far from sufficient to permit reproducibility.

Point-by-point query-response list to reviewers comments:

Reviewer #1:

Query#1: I think the order of the results and balance between the main results and supplementaries are bad. They first show the model of this study, followed by the SPR, structure and MD results. However, the model they showed first looks like summary of this study. Do they need to show the hypothesis first? Considering the variation of the conformation of domains, and the flexibility of substrates, it could be just two but many possibilities can be thought at the beginning of this study. Therefore, If we follow the “logic” of thinking, the model should be much later as a summary. I assume that unbalance situation is happened by the unbalance between main results and supplementaries. Since the data shown in this study are really massive, it should be many variations in the order of figures. Please consider the construction of the manuscript once again.

Response#1: We seriously considered the presentation order, but we chose to maintain the previous order as we believe it is more understandable for the reader especially in the light of the additional data included (see below). Moreover, the presentation and potential models are logical and in agreement with our previous published data (See ref 16 in the manuscript or PMID: 26610890).

Action#1: We have modified the text to ease clarity of the presentation flow.

Query#2: Is there any sugar moiety in the protein? Using Pichia pastoris as a host of recombinant enzyme production, it should have glycosylations. Isn't it affecting to the affinity and structure?

Response#2: The constructs encoding for GalNAc-T2 and T4 do not have N-glycosylation sites and are not glycosylated in Pichia pastoris. In addition, we did not observe any O-Man glycans either in our crystal structures.

Query#3: Please add line number for the reviewers convenience. Such considerateness is often important even the content is satisfactory, I think.

Action#3: Done

Query#4: Page 3, please add the information of family number for GT and possibly lectin domain.

Action#3: Done. See lines 77 and 79 in the manuscript where the CAZy GT27 and CBM13 are mentioned.

Query#4: Page 7, although they used bold font for the compounds, number only is rather difficult to distinguish figure number. Please use “Peptide 1”, "Compound 3” or so.

Action#4: Done

Query#5: Page 15, “O” should be italic

Action#5: Corrected

Query#6: Page 19, last sentence of first paragraph, How have the authors checked production of enzyme?

Response#6: Yes, production and purity of enzymes were evaluated by SDS-PAGE gels and BRADFORD protein assays. Pure enzymes (~95%) were quantified by absorbance at 280 nm using their theoretical extinction coefficient.

Action#6: We have included the following text on lines 507-509: “Purity of enzymes was evaluated by SDS-PAGE coomassie staining and quantification of enzymes was quantified by absorbance at 280 nm using their theoretical extinction coefficients”.

Reviewer #2:

Query#1: While the target has clearly some biomedical potential and is a viable and potentially impactful system to study conclusions about the lectin domain recognition and relative orientation with respect to the catalytic domain remain indirect. While the Xray structure only reveals a structural snapshot and STD NMR shows binding and orientation of the peptide other techniques might prove more convincing.

Response#1: We agree partially and believe that the revised manuscript with new studies and improved Figures address these concerns satisfactory. A new Fig. 3f shows superimposed structures of the two GalNAc-Ts with their substrates, and these structures clearly show the peptide acceptor regions aligned in the same orientation in the catalytic domain active site, while their N- or C-terminal prior GalNAc-O-Thr residues are bound to lectin domains positioned at vastly different locations relative to the catalytic domain. This clearly shows how the simple positioning the lectin domain can alter the long-range preferences of these transferases.

See below for other experiments we have tried to address the referee's comments. The composite data strongly indicate that the flexible linker is responsible for the rotational motion of the lectin domain.

Action#1: We have modified the text and now included the following additional data:

- 1) Additional kinetics and molecular dynamics studies (see Figures 2a, 4c, Supplementary Table 2 and Figure 6 and 8, and movies 1, 6 and 7).
- 2) New Figure 3f showing the superimposed structures of the catalytic domains of GalNAc-T2 and T4, bound to their preferred glycopeptide substrate.

Query#2: One could possibly study the domain orientations and flexibility of the linker in dependence of the mutants investigated by NMR PREs or single molecule fluorescence/FRET.

Response#2: While we fully agree with this, and we did make two attempts to make double mutants for both the WT enzyme and the chimera 2, we were unable to complete these experiments. Our idea was to use these mutants for PELDOR distance measurements to allow further demonstration that changes in the flexible linker results in the rotation of the lectin domain. The mutations for both the WT enzyme and chimera 2 were D259C-E472C and Q412C-N528C. All four of these residues are exposed to the solvent and do not interact with other residues in the structure. However, we could only express the double mutants corresponding to the WT enzyme preventing labeling of the mutants and the PELDOR distance experiments. The problem is that the chimeras express very poorly compared to the WT enzyme, and the double mutants in the chimera did not express as soluble secreted proteins in *Pichia pastoris*.

Our hypothesis was the following: In the compact structure of the WT GalNAc-T2, we would have expected to measure a distance of 31 Å for the double mutant Q412C-N528C in the presence of monoglycopeptide 2, whereas this distance in the chimera 2 and only in the presence of the monoglycopeptide 3, should be between 42-45 Å. A similar change would be expected for the other double mutant D259C-E472C. In this case, we would have a distance of 17 Å in the compact structure (assuming that is bound to the monoglycopeptide 2), that would change to 37 Å in chimera 2 (assuming that it is bound to the monoglycopeptide 3).

We also want to point out that the interpretation of these experiments if successful would not be straight forward, because as we showed earlier, GalNAc-T2 is in an equilibrium between compact and extended structures.

In summary, we were unable to perform additional NMR, FRET or PELDOR distance/dynamics experiments, and instead we present full kinetics of the WT GalNAc-T2 and T4, the chimera 3 and the triple mutant of GalNAc-T2. These experiments clearly support our main conclusions on the flexible linker being responsible of the rotational motion of the lectin domain. They clearly show that the swapping of the flexible linker or mutations in the flexible linker of GalNAc-T2 lead to different glycosylation preferences in these proteins and similar to the ones found for GalNAc-T4.

Action#2: Text has been modified to reflect the additional data introduced. We include the new data in Figures 2a and 4c.

Query#3: With the indirect evidence presented I struggle to see how these results will fundamentally change the field.

Response#3: We disagree strongly. Our data fully supports the hypothesis that the flexible linker of these enzymes dictates the rotational motion of their lectin domain, and thus is responsible for their distinct glycosylation preferences. This is the first time that this has been shown for this unique family of enzymes, and the first time anyone has explained how these enzymes could differentially recognize prior sites of glycosylation more than 10 residues from the acceptor site. As discussed in the manuscript this is likely an important feature for efficient glycosylation of regions with high density of O-glycans like in mucins and fundamental for our understanding of their biology.

Our previous report in the GalNAc-T2 structure bound to a glycopeptide was an important initial step towards this. However and only with the novel GalNAc-T4 structure reported here and the GalNAc-T2 chimeras and mutants, we demonstrate clearly the necessity for lectin domain movement, and the role of the linker domain in modulating this long-range specificity. We now present direct evidence that these enzymes have distinct glycosylation preferences because their GalNAc-binding site is differently located with respect to the catalytic domain. This in turn explains why they recognise the glycopeptides 2 and 3 in a different manner. Our data is supported by STD-NMR, X-ray crystallography, Molecular Dynamics simulations, and full kinetic characterization of chimeras and mutants. We believe that this is a major breakthrough in the field because we can now state that a small structural feature found in these enzymes, the flexible linker, is responsible of the translational and the rotational motion of the lectin domain. Finally, this provides the molecular basis of their distinct glycosylation preferences.

Reviewer #3:

Query#1: Experimentally, the best part of the study is the excellent X-ray crystal data and resulting structures. However, the structures and the information from them are not completely novel. E.g. some of the most important interacting residues have already been reported. Thus these structures seem more of a confirmative or supportive nature of what is already known.

Response#1: We disagree. We present for the first time a crystal structure of GalNAc-T4 bound to a glycopeptide, which provides an explanation for the previously observed inverse preference (N- vs C-terminal) for prior GalNAc-glycosites relative to GalNAc-T2. The key finding in this structure is the location of the GalNAc-binding site in the lectin domain. We show that is located 28 degrees with respect the same site in GalNAc-T2. This discovery is essential to explain why these enzymes have different long-range glycosylation preferences.

Action#1: We have modified and added a new Fig 3f to address and clarify this.

Query#2: The authors used three designed peptides, one un-glycosylated and two O-glycosylated to Thr residues (Table 1) to assess interactions and identify which part of the peptide/Thr is crucial to the interaction. However, their “best” peptide 3 with the N-terminal Thr-GalNAc is similar to that already reported.

Response#2: We are confused by this comment. The best peptide 3 to our knowledge has never been used on GalNAc-T4. In our earlier work we utilized a series of random glycopeptides to identify the long-range preferences of GalNAc-T4 (and other isoforms), and these would be unsuitable for our structural studies. We designed and used glycopeptide 3 as a tool to decipher the unique glycosylation preferences and structural binding properties of GalNAc-T4 in particular, based on published and unpublished data from our laboratory.

Action#2: None

Query#3: SPR was used to quantify peptide binding. Because binding of the peptides is very weak (high mM range), SPR is not the best technique to quantify such extremely weak binding interactions. Why were NMR titrations not performed? In NMR, interactions should be in the fast exchange regime on the chemical shift time scale, and accurate binding constants could likely have been determined.

Response#3: This is in principle correct. The proposed experiment would require ¹⁵N-labelled enzyme (and at least limited amide NH assignments), which was beyond the scope of this work. Usually, SPR-based affinity values take advantage of multivalent effects and the apparent binding is increased. We instead explored substrate binding by performing detailed enzyme kinetics and have now obtained the K_m value for the binding of glycopeptide 3 to GalNAc-T4. Contrary to the poor K_d , the K_m is of high affinity. The discrepancies between the high affinity K_m versus the poor K_d can be attributed to the fact that the kinetic characterisation is performed in the presence of UDP-GalNAc. It is known that this donor substrate stabilizes the flexible loop in a closed conformation, which is required to form part of the peptide-binding groove leading to an active enzyme (PMIDs: 24954443 and 25939779).

Action#3: We have modified the text and included new Fig4c and Supplementary Table 2.

Query#4: The STD NMR work provides some evidence for the binding epitope on the glycosylated peptides. However, STD effects are extremely small, with peptide 1 showing no effect, and peptides 2 and 3 showing only very weak effects, with most STD effects being around 1% or less and the maximum effect being around 3.5%. Moreover, STD NMR spectral baselines are not flat and thus likely contribute significantly to error, yet no statistical analyses were performed and values are shown to three significant figures. Therefore, these data are not fully convincing.

Response#4: We partly agree. From a quantitative perspective, the absolute STD values do look small. However, these are typical values for the weak nature of glycan/protein

interactions. We would like to emphasize that the beauty and power of STD reside in interpreting the relative STD values, which permits us to deduce the ligand binding epitope. We believe that the obtained perspective is clear and robust. We agree that we should not have emphasized the absolute values. Indeed, no STD response at all was obtained for the naked peptide. Additionally, the combined experiments indicate that GalNAc-T4 binds glycopeptides **2** and **3** through the lectin domain of the enzyme. It is also clear that the GalNAc protons receive much more saturation than the amino acid residues of the peptide backbone in glycopeptides **2** and **3**. The difference spectra are also very clear in this respect.

In any case, STD controls have also been taken into account to obtain the STD data displayed in Figure S1:

- i) Using a sample with only ligands, with no protein to remove the non-specific interactions due to non-specific saturation (this spectrum was previously considered in the first version to calculate the STD response).
- ii) Using a sample exclusively containing the protein to eliminate the deleterious baseline effects from the protein background (this spectrum was now taken in account in the actual version).

Action#4: We have included a revised Figure S1. The STD now displayed in account with the subtraction of both these two blank experiments.

Query#5: MD simulations have been run on the glyco-peptides and enzyme, and they appear to have been technically well done. However, the peptides bound to the enzyme interact in a highly dynamic fashion as stated by the authors (Fig 3e,f). In these instances, simulations have been run for 200 ns, but the stability of the complexes is unclear. No trajectories are shown and no binding energetics are provided. It could be that one needs a longer time frame to assess binding or changes in binding mode, yet no coarse grain simulations have been performed. More MD simulations (500 ns) are shown in Fig S6. However here it may be that rotations do not occur on that time scale. A longer time frame may be needed which coarse grain calculations would provide. Fig 4a also shows structures resulting from MD simulations. Once again, stability of the interactions with the peptides is unclear. For conclusions drawn solely from these MD simulations, experimental validation is needed.

Response#5: This is partly correct, but we believe that there is a misunderstanding regarding interpretation of the previous Fig. S6. This figure does not relate to the rotation of the lectin domain. This MD simulation was performed to show that the Thr acceptor sites were located close to the UDP, demonstrating that the lectin domain guides the delivery of acceptor sites into the catalytic domain.

The reviewers suggestion regarding the binding energetics is interesting, but the calculation of this term is not trivial using classical MD simulations, and we believe that this is beyond the scope of this study. The purpose of our MD simulation was to obtain a 3D picture of glycopeptide **3** bound to GalNAc-T4, and demonstrate the orientation of both GalNAc and the peptide fragment in the complex. Coarse-grain simulations, despite allowing extended simulation times, are not useful in this context since our main aim is to provide a detailed description of the (dynamic) recognition process at the atomic level (i.e. describe specific hydrogen bonds, hydrophobic interactions, etc. as shown for instance in Supplementary Figure 11).

We have performed a number of additional experiments as discussed in response#2 to reviewer #2, and new data is included.

Action#5: To clarify further we have included more detailed text of the MD simulations and a new Supplementary Figure 6, showing that the complex between GalNAc-T4 and glycopeptide **3** is stable during the entire simulation time. We have also included a Supplementary movie 1 of the trajectory.

We have also performed the following additional experiments:

a.-) We have made two sets of double mutants for both the WT enzyme and the chimera 2. These experiments were performed to address referee 2's comments in our proposed PELDOR distance experiments, which would have allowed us to further demonstrate that the flexible linker is responsible of the rotational motion of the lectin domain. These experiments unfortunately were largely unsuccessful due to the inability to express appropriate chimera mutant protein. Please see our reply to reviewer 2's second comments for a full discussion.

b.-) In the absence of the above experiments, we have performed full kinetics of the WT GalNAc-T2 and T4, the chimera 3 and the triple mutant of GalNAc-T2. These experiments clearly support our main conclusions on the flexible linker being responsible of the rotational motion of the lectin domain. They clearly show that the swapping of the flexible linker or mutations in the flexible linker of GalNAc-T2 lead to different glycosylation preferences in these proteins and similar to the ones found for GalNAc-T4. These experiments have been discussed in the text and shown in Fig. 4.

c)

Query#6: Table 2 shows a list of interesting and potentially informative enzyme linker-chimeras, yet only three could be expressed. Although limitations on expressing one protein or another is not unusual, it is unfortunate, as the whole set would have been nice to explore. The authors' results here suggest that enzyme chimera glycosylation preferences may be explained by rotation of the lectin domain as seen in MD simulations of chimera 2. However, once again, this is not definitive.

Response#6: We have now presented a large number of complementary experiments that further indicate that the flexible linker on these enzymes is responsible of their distinct glycosylation preferences. For example, the location of the GalNAc-binding site in GalNAc-T4 explains why this enzyme prefers to glycosylate monoglycopeptide **3** as shown in new Fig 3f. This is also supported by our full kinetics on the WT enzymes, the chimera 3 and the triple mutant.

Action#6: New data has been included as detailed above (see Figures 2a, 3f, 4c, Supplementary Table 2, Supplementary Figure 6 and 8, and movies 1, 6 and 7).

Query#7: Overall, the findings in this paper are generally interesting, yet mostly suggestive or supportive/confirmatory of what has been already reported. Thus the authors lack concrete proof for any single mechanism or validation of any one model already proposed model. For example, the GalNAc-T4 crystal structure indicates that the GalNAc-binding site of the lectin domain is rotated by 28° relative to the homologous GalNAc-T2 structure. Although the authors state that this provides “an explanation for the long-range glycosylation feature of GalNAc-T4”, it in fact only suggests that.

Response#7: We believe that the new Fig 3f clearly demonstrates how the rotation of the lectin domain leads to the changes in preferences. Moreover, the additional kinetic experiments conclusively demonstrate that the flexible linker is responsible for modulating the distinct glycosylation preferences observed.

Action#7: New data has been included as detailed above (see Figures 2a, 3f, 4c, Supplementary Table 2, Supplementary Figure 6 and 8, and movies 1, 6 and 7).

Query#8: And in terms of the flexible linker studies, the authors themselves state that their “flexible linker-swapping constructs of GalNAc-Ts with different glycosylation preferences (only) suggest that the flexible linker dictates rotation of the lectin domain”. In some ways, the findings presented here only provide an incremental advance to the field.

Response#8: We strongly disagree. Our findings do represent a major breakthrough in the understanding of the function of this large class of isoenzymes. We show that a small structural feature found in these enzymes - the flexible linkers - can lead to relatively dramatic changes in long-range preferences for glycosylated peptides. GalNAc-Ts are the only metazoan glycosyltransferase enzymes with lectin domains, and understanding their function will provide insight into how these enzymes orchestrate the complex process of coordinating O-glycosylation of thousands of proteins with quite diverse sequences and especially densities and patterns of O-glycan decoration. We provide a molecular basis, i.e. domain rotation/translation, for the observed distinct glycosylation preferences and our data agrees with a hierarchy model in which these isoenzymes have evolved to have different but complementary activities.

Reviewer #4:

Query#1: In my opinion the authors have shown that the differences in glycosylation patterns are "caused" by differences in the relative orientations of the lectin and catalytic domains. But whether or not the linker causes these orientational differences or merely allow them to occur remains unresolved. The authors repeatedly state that the linkers are flexible, as such it is unclear what might be the driving force that they could exert to cause the changes in domain orientation. In contrast if the linkers were rigid (or at least had strong conformational preferences), it is clear that they could directly control the domain orientations. The data from point mutations provide no clear support for a causative role for the linker.

Response#1: We agree that the generalized use of the term “flexible linker” may be misleading. By “flexible” we mean that the engineered linker must be to some extent able to recover its native low energy conformation after being artificially inserted in the computational model. The linker in GalNAc-T2 has a greater propensity to adopt a more folded conformation due to internal polar and hydrophobic interactions (see Supplementary Figure 8). Thus, when the linker of GalNAc-T3 is forced to adopt such folded conformation in the computational model for chimera 2, a significant structural stress develops in the linker; as a result, it effectively acts as a spring when allowed to relax early in the MD simulation (~30 ns, see Figure 4 and Supplementary Figure 8). As a result, and since protein-protein interactions between the lectin and catalytic domains in chimera 2 are not very intense, they quickly separate and the lectin domain subsequently rotates into the more relaxed orientation natively observed for GalNAc-T4.

We believe that the strong conformational preferences of the more folded native linker in GalNAc-T2 (see below for the implemented actions), dictate the unusual rotational state of the lectin domain in this enzyme, which can be overcome by linker engineering. This aspect also helps explain how GalNAc-T2 can display both N and C-terminal long-range preferences under certain substrate conditions.

In summary, we have unambiguously shown that this structural feature is responsible for the rotational/translational motion and in turn of the distinct glycosylation preferences found for GalNAc-Ts. Future studies would be needed to fully understand at the atomic level how the linker actually causes/promotes such lectin domain motion. [REDACTED]

[REDACTED]

[REDACTED]

Action#1: We have conducted additional MD simulations (500 ns) for the isolated flexible linkers of GalNAc-T2 and T3, and these studies are included in new Supplementary Movies 6 and 7. These show that while the GalNAc-T2 flexible linker adopts a more folded conformation, the GalNAc-T3 flexible linker adopts a more extended conformation. We have modified the text and included a sentence to qualify our usage of flexible linker on lines 274-278 of the manuscript (see Supplementary Movies 6 and 7)

Minor points that could enhance the manuscript would be:

Query#2: inclusion of a graph that plots the domain rotation as a function of the MD simulation time

Response#2: We agree.

Action#2: New Supplementary Figure 8 is included.

Query#3: Additional clarification of the manner in which the initial structures of the substrate-enzyme complexes were generated. The only information comes from one sentence "The peptide backbone of compound 3 was situated so that the Thr residues were laying on the proximity of the UDP moiety." This description is far from sufficient to permit reproducibility.

Response#3: We agree. We performed docking calculations using AutoDock Vina to generate the ternary complex as now fully described in the Methods of the revised manuscript. It is important to note that the lowest energy structure calculated for the glycopeptide do not locate the GalNAc unit in the lectin domain. Hence, the lowest energy structure that placed the sugar in this region was chosen to build up the initial 3D structure of the complex. These details are now included in the manuscript

Action#3: Additional text has been included in lines 625-641.

Reviewers' Comments:

Reviewer #1 (Remarks to the Author):

I think the manuscript has been revised intensively and now suitable for publication in Nature Communication.

Reviewer #3 (Remarks to the Author):

The authors have done an excellent job addressing reviewers' comments, adding new data, and modifying the manuscript overall. The paper now should be acceptable for publication in Nature Communications.

Reviewer #4 (Remarks to the Author):

The revised manuscript has addressed my concerns.